# Coupling of ATPase activity, microtubule binding, and mechanics in the dynein motor domain

Stefan Niekamp[1] , Nicolas Coudray[2,3], Nan Zhang[1], Ronald D Vale[1] & Gira Bhabha[2,*]

## Abstract

The movement of a molecular motor protein along a cytoskeletal track requires communication between enzymatic, polymer-binding, and mechanical elements. Such communication is particularly complex and not well understood in the dynein motor, an ATPase that is comprised of a ring of six AAA domains, a large mechanical element (linker) spanning over the ring, and a microtubule-binding domain (MTBD) that is separated from the AAA ring by a ~ 135 Å coiled-coil stalk. We identified mutations in the stalk that disrupt directional motion, have microtubule-independent hyperactive ATPase activity, and nucleotide-independent low affinity for microtubules. Cryo-electron microscopy structures of a mutant that uncouples ATPase activity from directional movement reveal that nucleotide-dependent conformational changes occur normally in one-half of the AAA ring, but are disrupted in the other half. The large-scale linker conformational change observed in the wild-type protein is also inhibited, revealing that this conformational change is not required for ATP hydrolysis. These results demonstrate an essential role of the stalk in regulating motor activity and coupling conformational changes across the two halves of the AAA ring.

**Keywords** cryo-electron microscopy; dynein; microtubule; motility; motor proteins
**Subject Categories** Cell Adhesion, Polarity & Cytoskeleton; Membrane & Intracellular Transport; Structural Biology
**The EMBO Journal (2019) 38: e101414**

## Introduction

Dyneins are minus-end directed, microtubule-based molecular motors that belong to the AAA+ (ATPases associated with diverse cellular activities) superfamily of proteins. Cytoplasmic dynein is responsible for the transport of numerous cargoes along microtubules (MTs), such as organelles, vesicles, viruses, and mRNAs (Vale, 2003; Vallee *et al*, 2004). In addition, cytoplasmic dynein plays key roles in facilitating basic cell biological processes such as

spindle positioning during mitosis (Kiyomitsu & Cheeseman, 2013). Mutations and defects in cytoplasmic dyneins are associated with many diseases such as neurodegenerative diseases and cancers (Roberts *et al*, 2013).

The cytoplasmic dynein holoenzyme is composed of two identical ~ 500 kDa heavy chains and multiple associated polypeptide chains that primarily bind to the N-terminal tail of dynein (Pfister *et al*, 2006). Regulatory proteins such as Lis1 and NudE bind to some dyneins and can modify its motility properties (Kardon & Vale, 2009; Vallee *et al*, 2012). To initiate processive motility for cargo transport, human cytoplasmic dynein also requires dynactin as well as cargo-adaptor proteins such as BicD and Hook3 (McKenney *et al*, 2014; Schlager *et al*, 2014). However, the core element for motility of all dyneins lies in the conserved motor domain of the heavy chain, which consists of six different AAA domains that are linked together as an asymmetric hexameric ring (AAA1–AAA6). Only AAA1–AAA4 can bind nucleotides (Burgess *et al*, 2003; Carter *et al*, 2011; Schmidt *et al*, 2015; Kon *et al*, 2012; Schmidt *et al*, 2012; Cho *et al*, 2008; Kon *et al*, 2004; Fig 1A); ATP hydrolysis in AAA1 is required for dynein stepping; and AAA3 acts as a switch that facilitates robust motility when ADP is bound (Bhabha *et al*, 2014, 2016; DeWitt *et al*, 2015). The catalytic domains in the AAA ring are spatially distant from the microtubule-binding domain (MTBD); the two are connected via the coiled-coil "stalk" that emerges from AAA4. Another coiled-coil element, called the buttress, protrudes from AAA5 and interacts with the stalk close to the ring (Fig 1A). The buttress also has been shown to be important for the allosteric communication between ring and MTBD (Kon *et al*, 2012). The N-terminal linker, which lies on top of the ring, is believed to serve as a mechanical element that drives motility (Burgess *et al*, 2003), and Can *et al* (2019) have recently shown that the direction in which the linker swings is critical to define the directionality of dynein. Over the last few years, several structural studies have illuminated a series of conformational changes in the dynein AAA ring during the ATPase cycle (Carter *et al*, 2011; Kon *et al*, 2012; Bhabha *et al*, 2014; Schmidt *et al*, 2015). The key conformational changes include domain rotations within the AAA ring and rearrangements of the linker domain.

To coordinate motility, motor proteins must communicate between the ATPase- and polymer-binding sites. ATP binding to AAA1 results in a weakened affinity ($K_d > 10$ μM) of dynein for

1 Department of Cellular and Molecular Pharmacology, Howard Hughes Medical Institute, University of California San Francisco, San Francisco, CA, USA
2 Department of Cell Biology, Skirball Institute of Biomolecular Medicine, New York University School of Medicine, New York, NY, USA
3 Applied Bioinformatics Laboratories, New York University School of Medicine, New York, NY, USA
*Corresponding author. Tel: +1 212 263 2959; E-mail: gira.bhabha@gmail.com

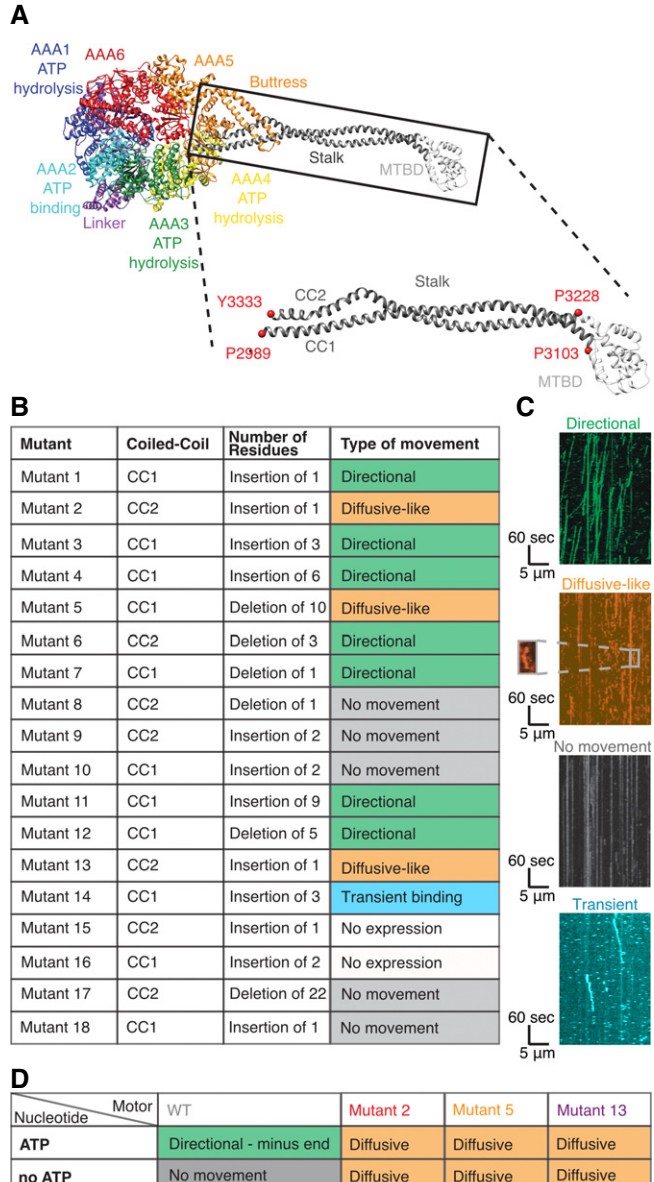

**Figure 1.** Single-molecule motility properties of dynein stalk mutants reveal mutants with nucleotide-independent diffusive motility.

A Structure and domain organization of the motor domain of cytoplasmic dynein [PDB 4RH7 (Schmidt *et al*, 2015)]. Inset shows zoom of MTBD (white) and coiled-coil stalk (gray), which consist of two helices, CC1 and CC2. Well conserved residues that were used as anchor points to define CC1 and CC2 in this study are depicted as red spheres. Numbering is based on yeast cytoplasmic dynein.

B Table showing location, number of inserted or deleted residues, and motility phenotype of all 18 stalk mutants. Examples of single-molecule assay results are shown in Fig EV2C and Appendix Fig S3. Sequence information and exact position of individual mutants are shown in Appendix Figs S1 and S2. Quantification and classification are based on three technical repetitions.

C Example kymographs for "Directional" (WT-like), "Diffusive-like", "No movement", and "transient binding". Magnified area for "Diffusive-like" motion shows run of a single molecule. Kymographs for each mutant are shown in Fig EV2C.

D Table showing the type of movement found for wild-type, mutant 2, mutant 5, and mutant 13 in a modified single-molecule assay with and without ATP. Classification of type of movement is based on two repetitions of different dynein preparations. Kymographs for each mutant are shown in Appendix Fig S4. Movies EV1–EV4 and EV5–EV8 show motility of wild-type and diffusive mutants with and without ATP, respectively.

To enable two-way communication between the MTBD and AAA ring, it has been suggested that the stalk undergoes conformational changes (Oiwa & Sakakibara, 2005; Shima *et al*, 2006; Kon *et al*, 2009). One hypothesis is that sliding between the two antiparallel helices of the stalk coiled-coil leads to changes in their register with respect to each other, with each registry corresponding to different microtubule affinities; the stalk in the β+ registry results in a low MT affinity state, and the α registry results in high MT affinity (Gibbons *et al*, 2005; Kon *et al*, 2009). This is further supported by structural work which has shown that when ADP-vanadate (ADP-vi) is bound to AAA1, the coiled-coil 2 (CC2) of the stalk is kinked and slides together with the buttress relative to coiled-coil 1 (CC1; Schmidt *et al*, 2015). Another study speculates that local melting of the coiled-coil between different states of the hydrolysis cycle plays a major role in the communication (Gee & Vallee, 1998; Nishikawa *et al*, 2016). However, how relative length changes in the stalk either via sliding or local melting drive the communication between the ring and the MTBD is not well understood.

To gain better insights into the allosteric communication between the AAA ring and the MTBD, we have identified mutants in the dynein stalk that block communication between the ATPase- and microtubule-binding sites. These mutants show diffusive movement along MTs and also hydrolyze ATP at maximal rates in a microtubule-independent manner. Structural characterization by cryo-electron microscopy (cryo-EM) of one of these mutants reveals a stabilization of a previously uncharacterized open conformation of the AAA ring in the presence of the non-hydrolyzable ATP analogue AMPPNP. In the presence of ADP-vanadate (ADP-vi), mimicking the post-hydrolysis state of dynein, we observed that this mutant is primed for hydrolysis, but with the linker in an extended conformation, which differs from the bent conformation of the linker in wild-type dynein (Bhabha *et al*, 2014; Schmidt *et al*, 2015). This result reveals that linker bending is not essential for ATP hydrolysis. Moreover, we gained new insights into domain movements in the AAA ring. The cryo-EM structure of the mutant in AMPPNP and

microtubules (MTs). After ATP hydrolysis and phosphate release, the motor binds MTs with stronger affinity ($K_d < 1$ μM; Kon *et al*, 2009). In this manner, the AAA ring controls the affinity of the MTBD for MTs. Conversely, interaction of the MTBD with MTs regulates the ATPase activity in the AAA ring (Kon *et al*, 2009). How this allosteric communication occurs is still poorly understood. In the case of kinesin and myosin, the ATPase- and track-binding sites are located relatively close (within ∼ 25 Å) to each other in the same domain (Vale & Milligan, 2000). In dynein, however, the very small ∼ 10 kDa microtubule-binding domain is spatially separated from the AAA ring by the ∼ 135 Å long coiled-coil stalk (Gibbons *et al*, 2005; Imamula *et al*, 2007; Carter *et al*, 2008; Kon *et al*, 2009; Redwine *et al*, 2012). Furthermore, the stalk is positioned between AAA4 and AAA5, which is on the opposite side of the ring from AAA1, resulting in a ∼ 240 Å separation between the main catalytic site and the MTBD.

ADP-vi states shows that one-half of the AAA ring undergoes a conformational change similar to the wild-type enzyme, while the AAA domain movements in the other half of the ring, from which the stalk extends, are disrupted. This result reveals that the stalk likely plays a key role in coupling conformational changes throughout the AAA ring. Our results provide insight into how conformational changes are coordinated within dynein's motor domain to allow microtubule regulation of ATPase activity and motility.

## Results

### Stalk mutants show nucleotide-independent diffusion

Given the spatial separation between dynein's catalytic AAA ring and the MTBD, it is apparent that allosteric communication must be mediated in some way via the stalk (Fig 1A). To understand what regions of the stalk may play a role in allosteric communication, we aligned and analyzed 534 sequences of dynein's motor domain. We found that the length of the stalk is very well conserved (99% of the sequences have the exact same stalk length) among species and types of dynein, such as cytoplasmic, axonemal, and IFT dynein, but the sequence is not (Appendix Note S1, Fig EV1). Based on the conserved length of the stalk and our sequence analyses, we decided to investigate how insertions and deletions in the stalk affect dynein's motility. We designed a panel of 18 insertion and deletion mutants in the yeast cytoplasmic dynein background, based on our sequence analysis (Fig 1B, Appendix Figs S1 and S2, Appendix Table S1). We expressed and purified GST-dimerized versions of each mutant (Fig EV2A) with an N-terminal GFP (Reck-Peterson *et al*, 2006; DeWitt *et al*, 2012), assessed the quality of the protein using negative stain electron microscopy to ensure structural integrity, and used single-molecule total internal reflection fluorescence (TIRF) microscopy assays (Reck-Peterson *et al*, 2006; Yildiz & Vale, 2015) for initial characterization of single-molecule motility.

Our panel of mutants displayed a wide variety of phenotypes (Fig 1B and C). Of the eighteen mutants, seven mutants (mutants 1, 3, 4, 6, 7, 11, and 12) showed single-molecule movement with velocities and processivity that were between ~ 50 and 100% of the wild-type protein (Fig EV2). Remarkably, some of these mutants had relatively large insertions of 6 (mutant 4) or 9 (mutant 11) residues or a deletion of five residues (mutant 12), but still moved in a similar way and direction as the wild-type motor. Of these seven mutants that showed wild-type phenotypes, six are in CC1 (mutants 1, 3, 4, 7, 11, and 12), suggesting that this helix is more tolerant of changes in length than CC2 (mutant 6; Fig EV1L, Appendix Figs S1–S3, Appendix Note S2). One region that is particularly sensitive to mutation is at the interface of the stalk and buttress (Fig EV1A). Most of the mutations that resulted in a dead (mutants 8, 9, 10, 17, and 18) or unstable (mutants 15 and 16) motor are clustered in the proximal region of the stalk, close to the AAA ring (Fig EV1L), and are in regions that are important for the stalk and buttress interaction (Fig EV1A and L, Appendix Fig S2). This stalk and buttress interface has been shown previously to play a role in nucleotide-dependent conformational change (Schmidt *et al*, 2015), and thus, our observations suggest that mutations in the stalk and buttress interface can severely compromise dynein motility (Appendix Note S2), consistent with the model that the stalk and buttress interface

is critical for dynein motility. Interestingly, we observed one mutant (mutant 14) that contains two distinct populations of molecules: The major population (96%, population 1) transiently binds to and releases from microtubules, and the minor population (4%, population 2) appears to move in a similar fashion as the wild-type motor (Fig EV3A and B). Surprisingly, the site of mutation for mutant 14 overlaps with that of mutant 18, yet single-molecule properties observed for mutant 18 show a dead motor (Fig EV2C, Appendix Figs S1 and S2). Lastly, three mutants from our panel (mutants 2, 5, and 13) presented a diffusive-like behavior, with single molecules randomly moving back-and-forth along the microtubule (Fig 1B and C, Appendix Fig S4, Movies EV1–EV4). This observation suggests that these diffusive-like motors are weakly bound to microtubules but unable to undergo effective unidirectional motion. We further analyzed the movement of mutant 5 along microtubules by measuring the displacement distance and directionality per 1-s interval. The histogram of the displacements (Appendix Fig S5) reveals a uniform Gaussian distribution centered close to zero with an average displacement of −3.3 nm. This analysis supported the notion that the back-and-forth motion of mutant 5 reflects random thermal-driven motion along the microtubule.

We decided to further characterize the three interesting mutants (mutants 2, 5, and 13) that showed similar one-dimensional diffusion along the microtubule and the one mutant (mutant 14) that showed weak binding and occasional directional motion. To assess the nucleotide dependence of the diffusive phenotypes, we carried out single-molecule experiments in the absence of ATP. As expected, the wild-type control showed no movement and was rigor bound to microtubules (Appendix Fig S4, Movie EV5). Surprisingly, in the absence of ATP, all three mutants (mutants 2, 5, and 13) displayed diffusive behavior very similar to that observed in the presence of ATP (Fig 1D, Appendix Fig S4). Diffusion, that we observed even in the absence of nucleotide, suggests that mutant 2, mutant 5, and mutant 13 have a weakened interaction with microtubules (Fig 1B, Appendix Fig S4, Movies EV5–EV8). Mutant 14 also seems to have weak affinity for microtubules in the apo state because we observed transient binding events in the absence of ATP (Fig EV3C).

We also assessed the nucleotide-dependent binding affinity of dynein for microtubules using a cosedimentation assay. In wild-type dynein, the motor binds tightly to microtubules in the absence of ATP, but weakly in the presence of ATP (Fig 2A). In contrast to the nucleotide-dependent microtubule affinity of wild-type enzyme, the microtubule affinity of the diffusive mutants (mutants 2, 5, and 13) and the transient binding mutant (mutant 14) was low in the absence of nucleotide and in the presence of ATP or AMPPNP (Figs 2B–D and EV3E, Appendix Table S2), which is consistent with the single-molecule motility results. These results confirm that the diffusive mutants and mutant 14 have a weakened microtubule affinity which remained unchanged in different nucleotide states.

Since we did not observe any directional movement of these three mutants in single-molecule assays, we asked whether there is any net directionality in a microtubule gliding assay when there are many motors interacting with a microtubule. In this microtubule gliding assay, dimeric dyneins (wild-type or mutants) were attached to a glass coverslip (Appendix Fig S6A). Results from this assay show that the three mutants generated microtubule gliding across the glass surface, although their velocities were ~ 10-fold lower than wild-type dynein (Appendix Fig S6B, Movies EV9–EV12). This

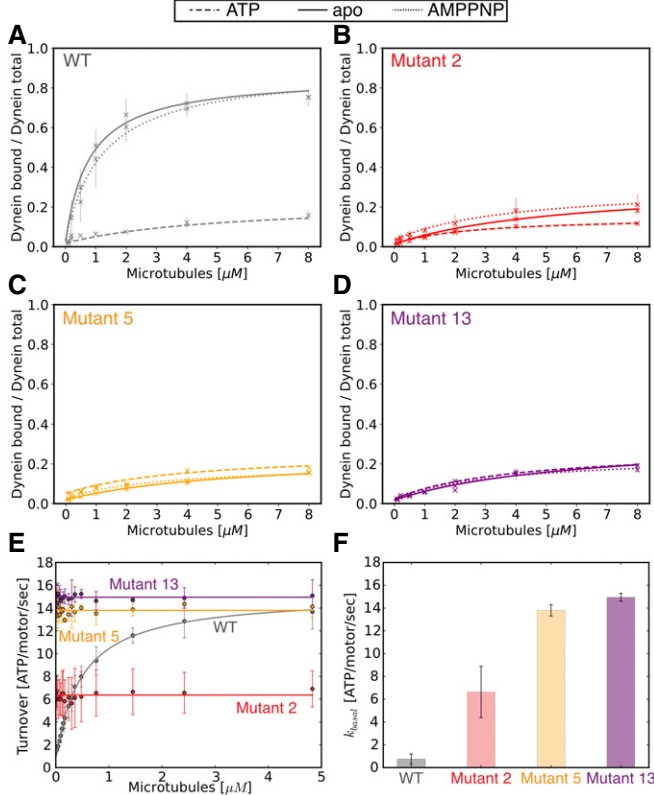

**Figure 2. Diffusive mutants show microtubule-independent, high basal ATPase activity, and low affinity for microtubules.**

A–D Microtubule affinity measured by a cosedimentation assay in the apo state (full line) and in the presence of ATP (dashed line), and AMPPNP (dotted line) for wild-type (A), mutant 2 (B), mutant 5 (C), and mutant 13 (D). Appendix Table S2 shows fit equation and rate quantification for microtubule affinity data.

E Microtubule-stimulated ATPase activity of wild-type (gray), mutant 2 (red), mutant 5 (orange), and mutant 13 (purple). Appendix Table S4 shows fit equation and rate quantification for ATPase data.

F Bar plot of basal ATPase activity of wild-type (gray), mutant 2 (red), mutant 5 (orange), and mutant 13 (purple).

Data information: Error bars show standard deviation of three repetitions of different dynein preparations.

phenotype is reminiscent of human cytoplasmic dynein (dynein 1) purified from rat brains (McKenney *et al*, 2014), which also shows diffusive motility in single-molecule assays, but shows robust directional movement in gliding assays. We also determined whether the microtubules were moving in the same direction as for wild-type dynein and assessed the direction of motion with single molecules of a human homodimeric kinesin 1 (K490; Tomishige *et al*, 2006), which moves processively toward the microtubule plus end (Appendix Fig S6A). By observing the direction of kinesin movement along the gliding microtubules, we could assess their polarity. Our results showed that the direction of mutant 2, 5, and 13 in microtubule gliding assays was the same as for wild-type dynein. In conclusion, mutants 2, 5, and 13 show nucleotide-independent diffusive movement as single molecules, while ensembles of these motors can produce extremely slow directional movement toward the MT minus end.

## Diffusive mutants show microtubule-independent hyperactive ATP hydrolysis

Since single-molecule analysis of mutants 2, 5, and 13 showed diffusive movement uncoupled from the nucleotide state, we asked whether these mutants were capable of hydrolyzing ATP. One possible hypothesis was that the mutants could no longer bind or hydrolyze ATP, while another possibility is that ATP hydrolysis was uncoupled from directional movement. We measured the ATPase activity of mutant 2, mutant 5, and mutant 13 at varying concentrations of microtubules. In wild-type dynein, ATPase activity is stimulated in the presence of microtubules, resulting in a characteristic increase in ATPase activity as microtubule concentration increased, until maximal ATPase activity is reached. For wild-type dynein, we measured a basal ATPase turnover of $0.75 \pm 0.34$ ATP/motor/s, which increased with increasing concentrations of microtubules to a $k_{\mathrm{cat}}$ of $15.18 \pm 1.18$ ATP/motor/s and $K_{\mathrm{M}}$ of $0.50 \pm 0.17$ µM for tubulin dimer (Fig 2E). These ATPase values are similar to those previously reported (Carter *et al*, 2008; Cho *et al*, 2008; Toropova *et al*, 2014; Appendix Table S3). Surprisingly, and in contrast to wild-type dynein, the three diffusive mutants (mutants 2, 5, and 13) showed high basal ATPase activity that did not significantly increase upon the addition of microtubules. Interestingly, the basal ATPase activities of mutants 5 and 13 were very similar to the maximal microtubule-stimulated ATPase activity of the wild-type protein (Fig 2E and F, Appendix Table S4). In addition to the diffusive mutants (mutants 2, 5, and 13), mutant 14 also showed high basal ATPase activity that is independent of microtubule concentration (Fig EV3D, Appendix Note S2). Together, these results indicate that the four weak binding mutants (2, 5, 13, and 14) all showed high ATPase activity and loss of microtubule regulation of the ATPase activity.

## Structural basis for hyperactivity of mutant 5

Our functional and biochemical assays showed that insertions and deletions in mutants 2, 5, and 13 result in (i) diffusive movement of single dynein molecules on microtubules, (ii) constitutively hyperactive ATPase, and (iii) constitutively weak microtubule binding that is not modulated by nucleotide. Taken together, these results suggest that these mutations disrupt the two-way communication between the MTBD and AAA ring in the dynein motor domain. Next, we sought to understand the structural basis underlying the uncoupling between the microtubule and ATPase sites in the diffusive mutants. Because of its high basal ATPase activity, we decided to focus on mutant 5.

We first collected a cryo-EM dataset of mutant 5 in the presence of 2 mM AMPPNP to mimic one of the post-force-generating states at AAA1 and AAA3. After 3D classification and refinement, we identified two distinct classes, with reconstructions at ∼ 7.5–8 Å resolution (Fig EV4A and B, Appendix Fig S7). This resolution allowed us to establish conformational changes at the subdomain level and model helices in some parts of the structure (Fig EV4C, Appendix Fig S7). Each AAA domain consists of a large subdomain (AAAL) and a small subdomain (AAAs), which can be considered as rigid bodies in the context of our resolution. Each AAAL and AAAs subdomain is fit independently as rigid bodies into each density map to generate a model corresponding to each map.

The most evident change in the motor domain of the majority (~ 71% of all particles—class 1, 7.7 Å resolution) of mutant 5 particles was a substantial opening between the small and large domains of AAA5 (Fig 3A and B), which was previously only observed as a minor conformation for the wild-type motor (Appendix Note S3, Appendix Fig S8). In addition, density for most of the distal stalk as well as the buttress is missing, suggesting that these regions are flexible. For the minor conformation (~ 29% of all particles—class 2, 7.6 Å resolution), the cryo-EM map shows a closed ring with no gap between the small and large domains of AAA5 and the helices of the initial part of the stalk and for the buttress are well defined (Figs 3A and EV4C). In this class, we can identify a conformation that has previously been referred to as the high microtubule affinity state in which the coiled-coil 2 of the stalk is not kinked (Fig EV4C; Schmidt *et al*, 2015). An additional and more subtle difference between the class 1 density and class 2 density is found in the N-terminal GFP tag at the end of the linker. In contrast to class 2, for class 1 (major class with "open" ring), the density for the N-terminal GFP tag is not well defined (Fig EV4D and E), which may indicate that the N-terminus of the linker is more flexible and potentially undocked from the ring at AAA5. Looking at domain movements in both class 1 and class 2 (Fig 3C and D), we also found that AAA2L is positioned away from the active site of AAA1 (Fig EV4F–R). Since the gap between AAA1 and AAA2 must close for productive ATP hydrolysis, we concluded that the ring of mutant 5 in the AMPPNP state is not primed for hydrolysis, as is true for wild-type dynein.

We next examined mutant 5 in the ADP-vanadate (ADP-vi) state, which mimics the post-hydrolysis state of dynein (Schmidt *et al*, 2015). In this state, the AAA domains in wild-type dynein adopt a more compact conformation in which the gap between AAA1 and AAA2 closes, which primes AAA1 for nucleotide hydrolysis (Fig EV4R). In addition, in the wild-type protein, the linker changes from a "straight" conformation (extended linker spanning from AAA1 to AAA5) to a "bent" conformation (the N-terminus of the linker making contacts with AAA 3/2). Our cryo-EM data for mutant 5 in the presence of 2 mM ATP and 2 mM vanadate resulted in a ~ 9 Å reconstruction, for which subdomain movements could be mapped with confidence (Figs 4A and EV5A). Based on fitting AAAs and AAAL domains into our density as described above, our data show that the gap between AAA1L and AAA2L for mutant 5 closes when transitioning from the AMPPNP to the ADP-vi state (Fig 4B, Movie EV13), similar to what was observed for wild-type dynein. The AAA2L domain of mutant 5 undergoes a rotation between the AMPPNP and ADP-vi state of ~ 21° which is similar to the ~ 20° domain rotation in wild-type dynein (Fig EV5C and D). The approximate distance between the arginine finger and Walker-A motif of mutant 5 and wild-type decreases from ~ 22 to ~ 20 Å in the AMPPNP state, respectively to ~ 17 and ~ 14 Å in the ADP-vi state, respectively (Fig EV5E), highlighting that the gap between AAA1L and AAA2L for mutant 5 and wild-type dynein indeed closes in a similar manner.

Unlike the bent linker observed for wild-type dynein in the presence of ADP-vi (Schmidt *et al*, 2015), the mutant 5 linker is not bent at the hinge-point (Fig 4A). However, the N-terminal region of the linker is undefined, suggesting increased flexibility at its N-terminal region (Figs 4A and C and EV5B). We confirmed the binding of

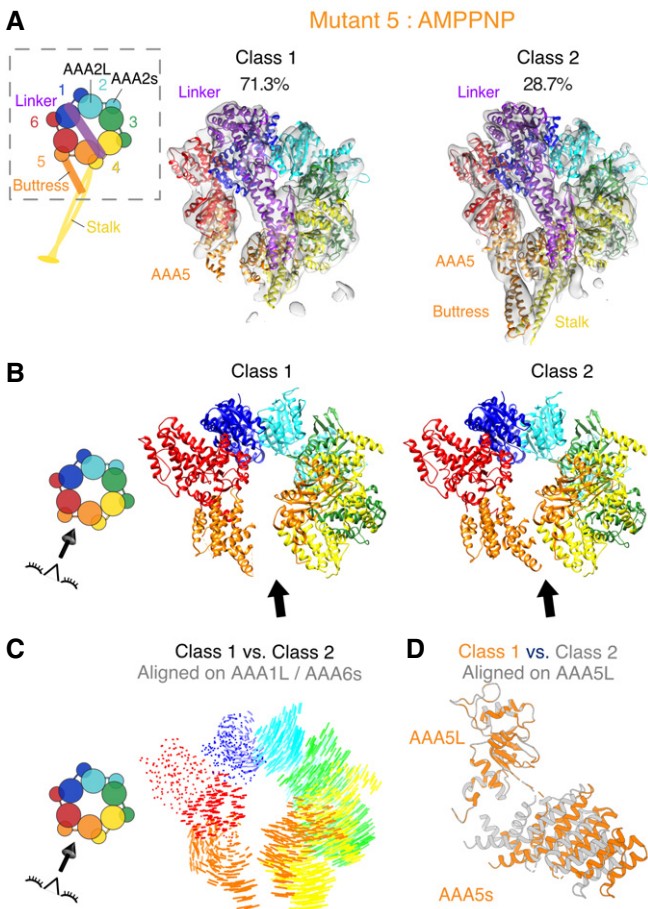

**Figure 3. Cryo-EM structure of mutant 5 in the presence of AMPPNP shows a gap in the AAA ring.**

A Cryo-EM reconstructions and fitted models for class 1 and class 2 resulting from 3D classification of the data. Class 1 is composed of 71.3% of all particles (left) and class 2 of 28.7% of all particles (right). The cryo-EM density map for both classes is shown as a semi-transparent surface with a fitted model (fit as described in Materials and Methods) as cartoon. Color coding of domains is the same as for Fig 1. Left: Schematic of monomeric dynein construct, box indicates region that was resolved in the cryo-EM maps.

B Cartoon representation of models for both classes. Black arrow indicates the position of the gap between AAA5L and AAA5s in class 1. Left: Schematic indicates the point-of-view.

C Visualization of interalpha carbon distances between class 1 and class 2 as shown in (B) after alignment on AAA1L and AAA6s. We removed the linker for clarity. Left: Schematic indicates the point-of-view.

D Movements between the large and small domains of AAA5 between class 1 (orange) and class 2 (gray). The large domain of AAA5 is aligned.

vanadate to the AAA1 nucleotide-binding pocket of mutant 5 by demonstrating vanadate-mediated UV photo-cleavage and vanadate inhibition of the ATPase activity (Fig EV5F–H). Thus, our structural data for mutant 5 in the presence of ATP and vanadate indicate that the motor is primed for hydrolysis, but does not undergo the large conformational change in the linker that is believed to be essential for motility (Burgess *et al*, 2003; Bhabha *et al*, 2016).

To better understand how mutant 5 can be primed for hydrolysis while the linker remains in a straight conformation, we analyzed the

AAA domain movements as dynein transitions from the AMPPNP to the ADP-vi state. When both states are aligned on AAA1L, we observe similar domain movements in approximately one-half of the ring surrounding AAA1 (from AAA5s to AAA2L; Fig 5A, Movie EV13), while the domain movements in the other half of the ring, AAA2s to AAA5L, are quite different (Figs 5B and EV5I). In contrast to the pronounced nucleotide-dependent motions in the AAA2s-AAA5L half of the ring for wild-type dynein, very little motion is observed for these domains in mutant 5 and the mode of movement is different (Fig 5B). Thus, between the AMPPNP and ADP-vi states, mutant 5 exhibits normal AAA domain movements in one-half of the ring (AAA5s-AAA2L), but shows a considerable lack of motion in the other half (AAA2s-AAA5L; Fig 5A and B). This result reveals that this stalk mutation uncouples nucleotide-dependent conformational changes in the two halves of the ring. Moreover, these results provide new insight into domain movements of the AAA ring and could explain why mutant 5 shows high ATPase activity but little motility, as will be described in the discussion.

## Discussion

We have identified mutations in the dynein stalk that show nucleotide-independent weak binding to microtubules and diffusional motion along the microtubule surface. A microtubule-stimulated ATPase assay revealed that these mutants hydrolyze ATP independently of microtubule concentration; two of these mutants (mutants 5 and 13) are hyperactive and have a basal ATPase activity that is as high as the maximal microtubule-stimulated turnover rate in the wild-type protein. Performing structural analysis on one of these mutants (mutant 5) using cryo-electron microscopy, we found that nucleotide-dependent "straight-to-bent" conformational change in the linker domain is inhibited. Moreover, we observed that AAA domain movements in one part of the ring are altered, while the other part of the ring becomes primed for hydrolysis very similarly as in wild-type dynein. These data provide new information on how the microtubule-binding domain (MTBD), stalk, linker, and AAA ring

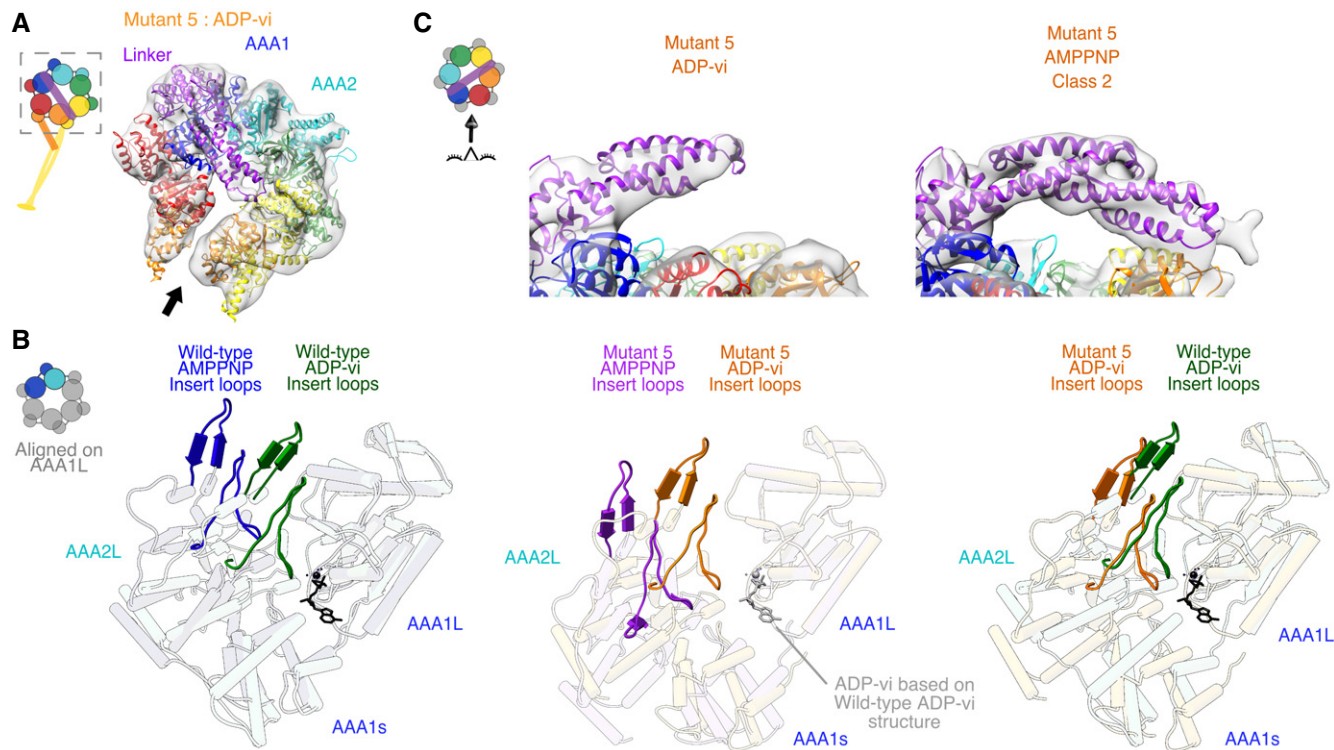

**Figure 4.  Cryo-EM structure of mutant 5 in the presence of ADP-vanadate shows priming for hydrolysis with unbent linker.**

A   Cryo-EM reconstructions and fitted models from 3D classification of the data. The cryo-EM density map is shown as a semi-transparent surface with a fitted model (fit as described in Materials and Methods) as cartoon. The black arrow indicates the position of the gap between AAA5L and AAA5s of mutant 5 in the presence of ADP-vi. This gap is smaller than for mutant 5 in the presence of AMPPNP but larger than in wild-type ADP-vi (Fig EV5J). Color coding of domains is the same as for Fig 1. Left: Schematic of monomeric dynein construct, box indicates region that was resolved in the cryo-EM map.

B   View of the AAA1 and AAA2 interface. The AAA2L inserts (the "H2 insert" and the "pre-sensor-I" (PS-I) insert) are shown in non-opaque colors. The structures of human cytoplasmic dynein 2 in the ADP-vi state [green—PDB: 4RH7 (Schmidt et al, 2015)], yeast cytoplasmic dynein in the AMPPNP state [blue—PDB: 4W8F (Bhabha et al, 2014)], yeast cytoplasmic dynein mutant 5 in ADP-vi state (orange—this study), and yeast cytoplasmic dynein mutant 5 in AMPPNP—class 1 state (purple—this study) were aligned on AAA1L. ADP and vanadate are depicted in black or gray and modeled based on the human cytoplasmic dynein 2 structure. Left: Schematic indicates region of nucleotide pocket. We also calculated the degree of rotation of AAA2L for the transition from the AMPPNP to the ADP-vi state of wild-type and mutant 5 and found rotations of 20° and 21°, respectively (Fig EV5C and D).

C   Close-up view of linker of cryo-EM reconstructions and fitted models for mutant 5 in ADP-vi (left) and in AMPPNP class 2 (right). For the ADP-vi state, only the part of the linker with sufficient density was fitted. Left: Schematic shows the point-of-view.

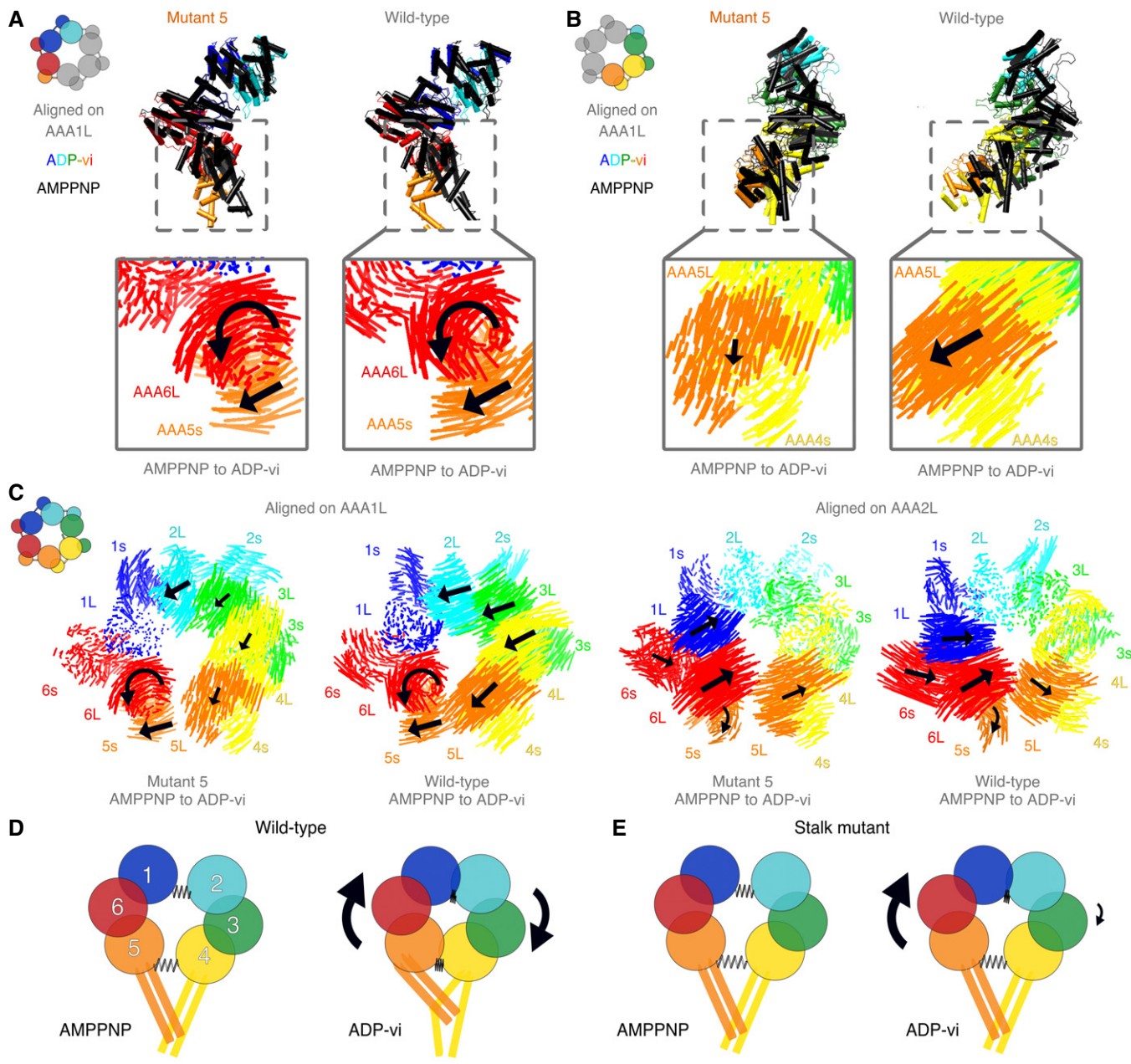

**Figure 5. Domain movements in the AAA ring of dynein.**

A   Domains AAA5s to AAA2L of wild-type and mutant 5 are shown for the ADP-vi state (color) and the AMPPNP state (black). Box: Visualization of interalpha carbon distances between AMPPNP and ADP-vi state of mutant 5 and wild-type dynein. Black arrows indicate direction of movement when transitioning from the AMPPNP to the ADP-vi state while the size of the arrow indicates the magnitude of movement. All structures are alignment on AAA1L. For wild-type, the structures of human cytoplasmic dynein 2 [ADP-vi state—PDB: 4RH7 (Schmidt *et al*, 2015)] and the yeast cytoplasmic dynein [AMPPNP state—PDB: 4W8F (Bhabha *et al*, 2014)] were used. For mutant 5 AMPPNP, we used the class 1 structure.

B   Same as in (A) but for domains AAA2s to AAA5L.

C   Visualization of interalpha carbon distances between the AAA wild-type domains in the AMPPNP and the ADP-vi state of mutant 5 and wild-type dynein for alignments on AAA1L (left) and AAA2L (right). Black arrows indicate direction of movement when transitioning from the AMPPNP to the ADP-vi state while the size of the arrow indicates the magnitude of movement. We removed the linker for clarity. For wild-type, the structures of human cytoplasmic dynein 2 [ADP-vi state—PDB: 4RH7 (Schmidt *et al*, 2015)] and the yeast cytoplasmic dynein [AMPPNP state—PDB: 4W8F (Bhabha *et al*, 2014)] were used. For mutant 5 AMPPNP, we used the class 1 structure.

D   Model for domain movements in the AAA ring of wild-type dynein during ATP hydrolysis at AAA1. The AAA ring can be divided into two halves that are connected by two springs. Upon ATP binding/hydrolysis, the gap between AAA1/AAA2 closes and moves AAA6/AAA5 which in turn pulls on AAA4 so that the gap between AAA4/AAA5 closes as well. In addition, this conformational change will pull the buttress and therewith change the stalk registry.

E   In the stalk mutant, the spring between AAA4/AAA5 does not close upon ATP binding/hydrolysis presumably due to a disruption of the stalk–buttress interface. Moreover, the gap at the AAA4/AAA5 interface is larger for the stalk mutant in both states, AMPPNP and ADP-vi, than for wild-type in the AMPPNP state. This "loose spring" at AAA4/AAA5 uncouples these domains from the closure of the AAA1/AAA2 interface, and this accounts for microtubule-independent hydrolysis.

communicate with one another during the ATPase cycle, as discussed below.

## Domain movements in the AAA ring

Dynein is a large and complex allosteric protein that must coordinate the conformations of four independent domains: (i) the AAA ring (consisting of 6 AAA domains), (ii) the largely helical linker (which spans over the ring and serves as a mechanical element), (iii) the small, globular microtubule-binding domain, and (iv) the stalk–buttress apparatus (a pair of antiparallel coiled coils that extend from the AAA ring and connect via the stalk to the microtubule-binding domain). Current structural data suggest that ATP binding to AAA1, with ADP bound at AAA3, drives full AAA ring closure (Kon *et al*, 2012; Bhabha *et al*, 2014; Schmidt *et al*, 2015), which is associated with a large-scale conformational change in the linker and a shift in registry of the two antiparallel coiled coils that affects the affinity of the distal microtubule-binding domain. However, the manner in which these different domains communicate with one another is incompletely understood.

Previous models for conformational changes in the AAA ring upon ATP binding suggest a rigid body movement of AAA2–AAA4, which propagates as a rotational motion to AAA5s and AAA6L which in turn pull the buttress relative to the stalk (Bhabha *et al*, 2014; Schmidt *et al*, 2015). ATP hydrolysis and/or product release then straightens the linker (thought to be the "power-stroke") and relaxes the ring back to its original conformation. Another model by Kon *et al* (2012) suggests an important role of the C-terminal domain (C-sequence), located on the surface of the ring opposite to the linker, in connecting AAA1L and AAA5s and triggering a movement of the buttress and, in turn, conformational changes in the stalk and MTBD. Even though the full C-terminal domain is not found in every dynein (Schmidt *et al*, 2015), the H1 alpha helix of the C-sequence that staples the AAA1L/AAA6s and AAA6L/AAA5s blocks together (Kon *et al*, 2012) appears to be present in virtually all dyneins.

Comparing the conformational states sampled by mutant 5 with the conformational states previously reported for the wild-type protein, we can understand how the stalk mutation in mutant 5 perturbs normal conformational changes in the dynein motor protein, thus provide insight into stalk-mediated conformational changes in the AAA ring.

Our cryo-EM data of mutant 5 revealed wild-type-like nucleotide-dependent AAA domain conformational rearrangements in one part of the ring (AAA5s-AAA6-AAA1-AAA2L), but absence/alteration of these conformational changes in the other part of the ring (AAA2s-AAA3-AAA4-AAA5L; Fig 5A and B, Movie EV13). Specifically, movements in AAA5L and AAA4s are much smaller in magnitude, and different in their vector of movement (Fig 5B, Movie EV13). This result indicates that the domain movements in the AAA5s-AAA2L block are insulated, at least to some extent, from the rest of the ring and from disruptive mutations in the stalk (Fig 5C, Movie EV14). Thus, we hypothesize that the two halves of the ring can undergo two independent modes of conformational change, which require the stalk–buttress apparatus to be properly coupled.

Based on the conformational changes seen in the cryo-EM data for mutant 5, we speculate that ATP binding does not primarily propagate in a clockwise (viewed from the linker side), domino-like

manner from AAA1 to AAA6 (Carter, 2013; Bhabha *et al*, 2014; Schmidt *et al*, 2015). Instead, the data for mutant 5 show bidirectional domain movement around the ATP-bound pocket of AAA1, with a block of AAA5s-AAA6Ls-AAA1L moving toward the AAA1s-AAA2L block upon ATP binding to AAA1 (Movie EV14). The C-terminal domain might provide the underlying bridging support between AAA5s/AAA6L and AAA6s/AAA1L that allows this block of AAA domains to move in a unified manner, consistent with the proposal of Kon *et al* (2012).

The structural data also emphasize the important role that the stalk–buttress plays in coupling the conformational changes in the two halves of the ring. Although we cannot see the precise lesion in the stalk caused by mutant 5 due to flexibility in this region, the downstream effect is an enlarged gap between AAA5s and AAA5L, which we speculate is the underlying cause in the disruption in the allosteric communication within the ring (Fig 5D and E). Specifically, our data for mutant 5 in the presence of ADP-vanadate suggest that movement of AAA5s/AAA6L toward the nucleotide-binding pocket of AAA1 is unable to pull AAA5L/AAA4s with it (and the other half of the ring; Movie EV14, Fig 5C). The interaction between the buttress (emerging from AAA5s) and stalk (emerging from AAA4s) is likely needed for this coordination between the two halves of the ring. Interestingly, the failure to connect AAA5L/AAA4s to the movement of the AAA5s-AAA6Ls-AAA1Ls-AAA2L block severely impacts the nucleotide-dependent movements of the half of the ring from AAA2s-AAA5L. This result suggests that domain movements in this part of the ring are dependent upon the integrity of the stalk–buttress apparatus and its connection to the autonomous nucleotide-driven motions of the AAA5s-AAA6Ls-AAA1Ls-AAA2L block.

This new model suggests why the stalk mutants hydrolyze ATP independently of microtubule concentration. Specifically, the disruption of the stalk–buttress interface allows the buttress and AAA5s/AAA6/AAA1L to undergo open-closed transitions accompanying ATP binding, hydrolysis, and product release, without any regulation by microtubules through the stalk–buttress apparatus (Fig 5D and E). Interestingly, buttress mutations that presumably disrupt the stalk–buttress interaction (Kon *et al*, 2012) also show high ATPase activity independent of microtubules, similar to our mutants 5 and 13. Thus, we suggest that the stalk–buttress interaction is a key regulator for dynein's ATPase activity by controlling the coupling between AAA4s/AAA5L and AAA5s/AAA6L and thus the coordination of domain movements in the two halves of the AAA ring.

## Uncoupling of linker bending from robust ATP hydrolysis

Our structural analysis revealed that mutant 5 has uncoupled nucleotide-dependent changes in one-half of the AAA ring and ATP hydrolysis from the large conformational change in the linker domain. Previous findings (Bhabha *et al*, 2014; DeWitt *et al*, 2015) suggested that in order for ATP hydrolysis to proceed, the linker must be undocked from the ring, to allow full closure of AAA2L. Thus far, structures in which the AAA ring is primed for hydrolysis (i.e., the gap between AAA2 and AAA1L is closed) have been accompanied by linker bending and docking onto AAA3/2 (Bhabha *et al*, 2014; Schmidt *et al*, 2015). However, our cryo-EM reconstruction of mutant 5 in the presence of ADP-vanadate shows a motor

that is primed for hydrolysis with an unbent linker (Fig 4A and C). Weak/missing density for the N-terminus of the linker suggests that it is flexible and might be undocked; however, it is clearly not in a bent conformation (Fig EV5B). These data therefore suggest that linker bending is not a prerequisite to prime the motor for hydrolysis. However, the linker bending is not only necessary for efficient directional motion (Bhabha *et al*, 2014; Cleary *et al*, 2014; since single mutant 5 motors show random bidirectional motion) but also important to set the directionality of the motor (Can *et al*, 2019). Together, these data further support our model that the hydrolysis cycle arises from autonomous conformational changes within the AAA5s-AAA6Ls-AAA1Ls-AAA2L block and neither require linker bending nor an intact stalk–buttress interface.

Our structural data also allow us to speculate on how linker bending is initiated and why it fails to occur in the mutant 5. In the nucleotide-free and ADP state, the linker forms contacts with AAA5L. Movement of AAA5L upon ATP binding in AAA1 may induce a steric clash with the linker [perhaps with contributions from AAA4L (Schmidt *et al*, 2015)], and linker bending may ensue to minimize such clashes. In mutant 5 however, AAA5L movement is minimal and thus may be unable to induce the linker steric clash (Fig EV5K), which is highlighted in the "gap" between AAA5L and AAA5s. We therefore speculate that the interface between AAA5s and AAA5L (AAA5s connecting to the AAA1 ATPase site via the AAA5s-AAA6Ls-AAA1Ls-AAA2L block and AAA5L interacting with the linker and AAA4s/stalk/MTBD) may be a critical region for coordinating ATPase activity (AAA1), microtubule binding (stalk/MTBD), and mechanics (linker) in the dynein motor domain (Movie EV14).

# Materials and Methods

## Bioinformatic analysis of dynein sequences

The detailed process is described in Appendix Note S1. Briefly, we used 677 unique axonemal and cytoplasmic dynein heavy chain sequences from 229 fully sequenced eukaryotic genomes, which we received from Christian Zmasek, Godzik Lab, Burnham. This dataset was pruned based on well-defined criteria as listed in Appendix Note S1 and analyzed using Jalview (Waterhouse *et al*, 2009). Remaining sequences were aligned using MAFFT (Katoh *et al*, 2002) in the Bioinformatic Toolkit (Alva *et al*, 2016), and mutations in the stalk were manually identified by comparing sequences in Jalview. More information on the sequence alignments can be found in the Appendix Note S1.

## Yeast strains used in this study

Recombinant *Saccharomyces cerevisiae* cytoplasmic dynein (Dyn1) truncated at the N-terminus (1219–4093 aa) was used in this study. All constructs used in this study are listed in Appendix Table S1. Dimeric constructs are based on VY208 and were created by artificially dimerization through an N-terminal GST-tag (Reck-Peterson *et al*, 2006) and tagged with a HaloTag (Promega) at the C-terminus as well as a GFP at the very N-terminus. Monomeric constructs (VY137) are GFP tagged at the N-terminus. Stalk mutations were inserted by homologous recombination as previously described (Reck-Peterson *et al*, 2006).

## Protein expression and purification

Dynein was expressed and purified as previously described (Reck-Peterson *et al*, 2006). Monomeric and dimeric constructs were further purified by gel filtration on a GE Healthcare Superdex 200 10/300GL and a GE Healthcare Superose 6 10/300GL column, respectively, in dynein gel filtration buffer (50 mM K-Ac, 20 mM Tris, pH 8.0, 2 mM Mg(Ac)$_2$, 1 mM EGTA, 1 mM TCEP, and 10% glycerol) and flash-frozen afterward. The "cysteine-light" human ubiquitous kinesin 1 dimer E215C K490 construct was cloned and purified as previously described (Tomishige *et al*, 2006; Mori *et al*, 2007). Following dialysis, the E215C K490 construct was reacted for 4 h at 4°C with Cy3-maleimide (GE Healthcare, PA13131) at a motor/Cy3 dye ratio of 1:10 as previously described (Tomishige *et al*, 2006). The unreacted maleimide dyes were then quenched with 1 mM dithiothreitol (DTT). Afterward, the kinesin was purified by gel filtration over a S200 10/300GL column (GE Healthcare) in kinesin gel filtration buffer (25 mM Pipes (pH 6.8), 2 mM MgCl$_2$, 200 mM NaCl, 1 mM EGTA, 1 mM DTT, and 10% sucrose) and then flash-frozen.

## Microtubule preparation

Tubulin was purified and polymerized as previously described (McKenney *et al*, 2014). For single-molecule motility assays, unlabeled tubulin, biotinylated tubulin, and fluorescent tubulin were mixed at an approximate ratio of 20:2:1 in BRB80 (80 mM Pipes (pH 6.8), 1 mM EGTA, and 1 mM MgCl$_2$). For the gliding assay, unlabeled tubulin and fluorescent tubulin were mixed at an approximate ratio of 20:1 in BRB80. For tubulin that was used in the ATPase assay as well as the microtubule affinity assay, only unlabeled tubulin was used. We added 1 mM GTP to all polymerization reactions. Then, the mixtures were incubated for 15 min in a 37°C water bath. 20 μM of Taxol (Sigma, T1912) was added afterward, and the mixture was incubated for 2 more hours at 37°C. Before usage, microtubules were spun over a 25% sucrose cushion in BRB80 at ~160,000 *g* for 10 min in a tabletop centrifuge.

## Gliding and single-molecule motility assay

We made custom flow chambers using laser-cut double-sided adhesive sheets (Soles2dance, 9474-08x12—3M 9474LE 300LSE). We used glass slides (Thermo Fisher Scientific, 12-550-123) and coverslips (Zeiss, 474030-9000-000). We cleaned the coverslips in a 5% v/v solution of Hellmanex III (Sigma, Z805939-1EA) at 50°C overnight and then washed them extensively with Milli-Q water. The flow cells were assembled in a way that each chamber holds approximately 10 μl.

Every data collection was carried out at room temperature (~ 23°C) using a total internal reflection fluorescence (TIRF) inverted microscope (Nikon Eclipse Ti microscope) equipped with a 100 × (1.45 NA) oil objective (Nikon, Plan Apo λ). We used an Andor iXon 512 × 512 pixel EM camera, DU-897E and a pixel size of 159 nm. Dynein [always as dimer and either labeled with GFP only or with GFP and a Halo488 dye (Promega, G1001)] was excited with a 488 nm laser (Coherent Sapphire 488 LP, 150 mW), kinesin with a 561 nm laser (Coherent Sapphire 561 LP, 150 mW), and microtubules with a 640 nm laser (Coherent CUBE 640-100C,

100 mW). For the gliding assay, images were recorded with 100-ms exposure time and a 2-s frame rate for MTs and a 100-ms frame rate for kinesin. For the single-molecule assay of dynein, we used 100-ms exposures and a 2-s frame rate and a 100-ms frame rate for kinesin. The acquisition software was µManager (Edelstein *et al*, 2010) 2.0, and data were analyzed in ImageJ (Schneider *et al*, 2012).

For the gliding assay, we first added 10 µl of GFP antibody (Abcam, ab1218) and incubated for 5 min. Then, we washed with 20 µl of DAB with 2 mg/ml β-casein and 0.4 mg/ml κ-casein. We then added 10 µl of dimeric dynein and incubated for another 5 min which was followed by an additional wash with 20 µl of DAB with 2 mg/ml β-casein and 0.4 mg/ml κ-casein. Next, we added 10 µl of polymerized microtubules and incubated for 5 min. Then, we washed with 30 µl of DAB with 2 mg/ml β-casein and 0.4 mg/ml κ-casein. Finally, 10 µl of DAB with kinesin, 0.4 mg/ml κ-casein, 10 µM Taxol, 1 mM Mg-ATP, and the PCA/PCD/Trolox oxygen scavenging system (Aitken *et al*, 2008) were added.

Prior to the single-molecule motility assays, dynein was labeled with Halo488 dye (Promega, G1001) as previously described (Bhabha *et al*, 2014). Briefly, dynein constructs were mixed with 20 µM Halo Alexa488 dye and incubated on ice for 10 min and a PD MiniTrap G-25 column (GE Healthcare) equilibrated with dynein gel filtration buffer was used to remove excess dye afterward.

The flow chambers for the single-molecule motility assay were prepared as previously described (Yildiz & Vale, 2015). Briefly, we first added 10 µl of 5 mg/ml Biotin-BSA in BRB80 and incubated for 2 min. Then, we washed with 20 µl of BRB80 with 2 mg/ml β-casein (Sigma, C6905) and 0.4 mg/ml κ-casein (Sigma, C0406). Afterward, we added 10 µl of 0.5 mg/ml streptavidin in PBS for a 2-min incubation. Next, we again washed with 20 µl of BRB80 with 2 mg/ml β-casein and 0.4 mg/ml κ-casein. This was followed by the addition of 10 µl of polymerized microtubules and a 5-min incubation. Then, we washed with 30 µl of DAB (50 mM K-Ac, 30 mM HEPES, pH 7.4, 2 mM Mg(Ac)₂, 1 mM EGTA) with 2 mg/ml β-casein, 0.4 mg/ml κ-casein, and 10 µM Taxol. Finally, we added 10 µl of dynein and kinesin in DAB with 0.4 mg/ml κ-casein, 10 µM Taxol, 1 mM Mg-ATP, and the PCA/PCD/Trolox oxygen scavenging system (Aitken *et al*, 2008). In the single-molecule assay where ATP was omitted, the final solution contained 10 µl of dynein in DAB with 0.4 mg/ml κ-casein, 10 µM Taxol, and the PCA/PCD/Trolox oxygen scavenging system (Aitken *et al*, 2008). The acquisition software was µManager (Edelstein *et al*, 2010) 2.0.

### Diffusion analysis of single-molecule movements

The imaging was performed as described for the single-molecule motility assay. Subsequently, tracks of wild-type dynein and mutant 5 along microtubules in the presence of 1 mM ATP were obtained. We tracked single molecules using the "localization microscopy" plug-in from µManager 2.0 (Edelstein *et al*, 2010) by fitting emitters with a Gaussian-based maximum-likelihood estimation (Mortensen *et al*, 2010) as previously described (Niekamp *et al*, 2019) and extracted tracks based on a nearest neighbor search. We then straightened these single-molecule traces along the main axis of motion along the microtubule using a principal component analysis implemented in the "localization microscopy" plug-in from

µManager 2.0 (Edelstein *et al*, 2010). Following this, the displacement of wild-type and mutant 5 was binned into 1-s intervals and the polarity of microtubules was determined by analyzing the directionality of human homodimeric kinesin 1 (K490; Tomishige *et al*, 2006), which moves processively toward the plus end of microtubules.

### ATPase assay

The ATPase assays were carried out in DAB (50 mM K-Ac, 30 mM HEPES, pH 7.4, 2 mM Mg(Ac)₂, 1 mM EGTA) as follows. We mixed dynein (monomeric for all constructs) to a final concentration of 10–20 nM with 2 mM Mg-ATP (Sigma), 0.2 mM NADH (Sigma), 1 mM phosphoenolpyruvate (Sigma), 0.01 U pyruvate kinase (Sigma), 0.03 U lactate dehydrogenase (Sigma), 10 µM Taxol, 1 mM DTT, and 0–5 µM microtubules in DAB. Absorbance at 340 nm was continuously measured in an Eppendorf Spectrophotometer (UV-Vis BioSpectrometer), and the data were fit to the following equation (Bhabha *et al*, 2014) using an excel curve fitting routine:

$$k_{\mathrm{obs}} = (k_{\mathrm{cat}} - k_{\mathrm{basal}}) \frac{[\mathrm{MT}]}{K_{\mathrm{M}} + [\mathrm{MT}]} + k_{\mathrm{basal}}.$$

The vanadate inhibition of dynein ATPase activity was performed as previously described (Höök *et al*, 2005). Briefly, we mixed dynein [monomeric of wild-type and mutant 5 (from the same batch that was used to solve the structure of mutant 5 in the presence of ADP-vi)] to a final concentration of 20 nM with 1 mM Mg-ATP (Sigma), 0.2 mM NADH (Sigma), 1 mM phosphoenolpyruvate (Sigma), 0.01 U pyruvate kinase (Sigma), 0.03 U lactate dehydrogenase (Sigma), 10 µM Taxol, 1 mM DTT, 6 µM microtubules, and 0–100 µM vanadate (Sigma) in DAB. The vanadate was boiled for 10 min before usage. Absorbance at 340 nm was continuously measured in an Eppendorf Spectrophotometer (UV-Vis BioSpectrometer), and the turnover rate was calculated as described above.

### Vanadate-mediated UV photo-cleavage

Protein from the same batch that was used to solve the structure of mutant 5 in the presence of ADP-Vi was used in the vanadate-mediated UV photo-cleavage. The assay was performed in a similar way as previously described (Schmidt *et al*, 2015). Briefly, mutant 5 monomer was mixed with 2 mM Mg-ATP (Sigma) and 2 mM vanadate (Sigma) and either exposed to UV-light (365 nm) or kept in the dark for 90 min. The vanadate was boiled for 10 min before usage. Afterward, the samples were analyzed by SDS–PAGE.

### Microtubule affinity assay

The microtubule affinity assays were carried out in DAB (50 mM K-Ac, 30 mM HEPES, pH 7.4, 2 mM Mg(Ac)₂, 1 mM EGTA) as follows. We mixed dynein (monomeric for all constructs) to a final concentration of approx. 50 nM with 10 µM Taxol, 1 mM DTT, and 0–8 µM microtubules in DAB. For the measurements with ATP, we added 5 mM Mg-ATP (Sigma), and for the experiment with AMPPNP, we added 5 mM Mg-AMPPNP (Sigma). After a 3-min incubation at room temperature, the samples were spun over a 25 %

sucrose cushion in DAB at ~160,000 *g* for 10 min in a tabletop centrifuge. The concentration of dynein in the supernatant (unbound) and in the pellet (bound) was determined by measuring the intensity of the N-terminal GFP on a Typhoon laser scanner (GE Healthcare). The data were fit to the following equation using an excel curve fitting routine $k_{\mathrm{obs}} = (B_{\mathrm{M}} - k_{\mathrm{basal}})\frac{[\mathrm{MT}]}{K_{\mathrm{d}}+[\mathrm{MT}]} + k_{\mathrm{basal}}$ in which $B_{\mathrm{M}}$ is the maximum binding, $K_{\mathrm{d}}$ is the dissociation constant, $k_{\mathrm{basal}}$ is the basal "binding" fraction and accounts for the pelleting of dynein without microtubules present, and $k_{\mathrm{obs}}$ is the observed fraction of dynein bound (pelleted) over the total amount of dynein. We could have also used the simplified equation $k_{\mathrm{obs}} = (B_{\mathrm{M}})\frac{[\mathrm{MT}]}{K_{\mathrm{d}}+[\mathrm{MT}]}$ ($B_{\mathrm{M}}$ maximum binding, $K_{\mathrm{d}}$ dissociation constant) but we wanted to account for potential pelleting of dynein without microtubules ($k_{\mathrm{basal}}$). However, since $k_{\mathrm{basal}}$ is very low, using the simplified equation gives almost identical results for $B_{\mathrm{M}}$ and $K_{\mathrm{d}}$.

**Electron microscopy data collection**

For negative stain, data for mutants 5 (monomer) were collected on a Tecnai F20 microscope with a Tietz F416 CMOS detector at the New York Structural Biology Center (NYSBC). Leginon software (Suloway *et al*, 2005) was used for the semi-automated collection of 825 images at a magnification of ×62,000 and a pixel size of 3 Å per pixel. For cryo-EM data collection, 1,200 movies of mutant 5 (monomer) mixed with 2 mM AMPPNP were recorded with SerialEM (Mastronarde, 2005) at 300 kV on a Titan Krios (FEI) equipped with a K2 summit camera (Gatan) at 0.655 Å per pixel in super-resolution mode at Janelia Research Campus. Another 664 movies of the same mutant (mutant 5—monomer) mixed with 2 mM ATP and 2 mM vanadate were recorded with SerialEM at 200 kV on a Arctica (FEI) equipped with a K2 summit camera (Gatan) at 0.578 Å per pixel in super-resolution mode at New York University.

**Electron microscopy data processing and analysis**

For the images of the negatively stained sample, particles were selected using DoG picker (Voss *et al*, 2009) in APPION (Lander *et al*, 2009) and then extracted in Relion 2.1.0 (Scheres, 2012) into boxes of 180 × 180 pixels, leading to 156,199 boxes for mutant 5. A round of 2D classification was performed to remove junk and noisy particles, leading to 54,913 particles selected. Subsequent image processing steps were carried out using CryoSPARC (Punjani *et al*, 2017). After having generated an *ab-initio* model, those particles were used to generate eight 3D classes. Because of the similarity between all those classes, a final round of 3D refinement was completed using all of the particles.

For the cryo-EM images (see also Appendix Table S5), the movies of mutant 5 with 2 mM AMPPNP were first aligned and binned to 1.31 Å per pixel with MotionCor2 v1.0.5 (Zheng *et al*, 2017), and then, the contrast transfer function parameters were estimated with GTCF 1.06 (Zhang, 2016). The particles were picked automatically in Relion 2.1.0 (Scheres, 2012) using a Gaussian blob as a reference, and further processing was done in CryoSPARC (Punjani *et al*, 2017). Out of the 310,085 regions

automatically picked, 136,056 were kept after evaluation of 2D classes. Two *ab-initio* models were first generated in CryoSPARC, and the best one was used in a 4-class 3D heterogeneous refinement. Then, two 3D homogeneous refinements were completed: one with class 3 (here referred to as class 2—with 29% of remaining particles and with a resolution of 7.6 Å) and another one (here referred to as class 1—with 71% of remaining particles and with a resolution of 7.7 Å) with the three other classes which looked very similar and were therefore combined before refinement. Note that the overall and local resolutions we report are the average resolution after refinement in CryoSPARC v2.5.0 (Fig EV4A and B, and Appendix Fig S7). The final maps were then filtered for display using a B-factor of −400. For modeling, we used PDB 4W8F as a reference. The PDB file was split into 13 domains (small and large subdomains for each AAA domain, and the linker), and for each of those domains, we simultaneously fit all 13 subdomains into the map using UCSF Chimera (Pettersen *et al*, 2004). We noticed that the rigid body of the buttress region in class 2 map did not perfectly fit the densities (Fig EV4C). This model was therefore subjected to the real_space_refine algorithm in PHENIX (Adams *et al*, 2010) using two cycles and 100 iterations to optimize the fit. Figures and movies were generated with the UCSF Chimera package or the Pymol Molecular Graphics System (version 2.0, Schröodinger, LLC).

For the images of mutant 5 with 2 mM ATP and 2 mM vanadate acquired on the Arctica, a similar process was followed. First aligned and binned to 1.31 Å per pixel with MotionCor2 v1.0.5 (Zheng *et al*, 2017), and the contrast transfer function parameters estimated with GCTF 1.06 (Zhang, 2016). A first round of auto-picking was conducted in Relion 2.1.0 (Scheres, 2012) using a Gaussian blob as a reference. Two of the resulting classes were then used as template for a round of reference-based auto-picking. Further processing was also conducted in CryoSPARC (Punjani *et al*, 2017). Out of 35,565 picked particles, 32,442 particles were kept for the generation of ab-initio models and a 4 and a 5-class heterogeneous refinement were tried. One class with 8,653 particles leads to a clear dynein 3D model that we refined to 9.2 Å (Fig EV5A) and finally filtered for display using a B-factor of −400. Another class at 17 Å seemed to show only the AAA domains while the linker could not be seen. The model for the 9.2 Å map was constructed as for mutant 5 in the AMPPNP state, using rigid body docking of domains from PDB 4W8F, but was not further refined in PHENIX due to its lower resolution.

**Figure preparation**

Figures and graphs were created using Pymol (version 2.0 Schrödinger, LLC) and Chimera (Pettersen *et al*, 2004; structure representation), ImageJ (Schneider *et al*, 2012; light microscopy data), Jalview (Waterhouse *et al*, 2009; sequence analysis and representation), Affinity designer (version 1.6.1, Serif (Europe) Ltd), and Python (version 2.7, Python Software Foundation).

**Statistics**

For each result obtained, the inherent uncertainty due to random or systematic errors and their validation are discussed in the relevant sections of the manuscript. Details about the sample size, number of

independent calculations, and the determination of error bars in plots are included in the figures and figure captions.

## Code availability

μManager acquisition and analysis software are available partly under the Berkeley Software Distribution (BSD) license and partly under the GNU Lesser General Public License (LGPL), and development is hosted on GitHub at https://github.com/nicost/micro-manager. The latest version for MacOS and Windows can be downloaded here: https://valelab.ucsf.edu/ ~ nico/mm2gamma/.

## Data availability

Density maps for the structures were deposited in the Electron Microscopy Data Bank under accession codes EMD-7829 (mutant 5 in the presence of AMPPNP, class 1), EMD-7830 (mutant 5 in the presence of AMPPNP, class 2), and EMD-9386 (mutant 5 in the presence of ADP-vanadate). The sequence alignment files used to create the mutants are available as Appendix. All other data are available from the corresponding author upon request.

**Expanded View** for this article is available online.

## Acknowledgements

We are grateful to J. Sheu-Gruttadauria, Iris Grossman-Haham, and Zhen Chen for critical discussions of the article. We would like to thank Christian Zmasek at the Burnham Institute for the initial sequence alignment file. We thank Nico Stuurman and Walter Huynh for their assistance and advice in light microscopy. Some of this work was performed at the Simons Electron Microscopy Center and National Resource for Automated Molecular Microscopy located at the New York Structural Biology Center, supported by grants from the Simons Foundation (349247), NYSTAR, and the NIH National Institute of General Medical Sciences (GM103310). We thank Kelsey Jordan at the New York Structural Biology Center for assistance with data collection of negatively stained samples. Cryo-EM data were collected on the Titan Krios ("Krios 2") at Janelia Research Campus and the Talos Arctica at the NYU Langone Health's Cryo−Electron Microscopy Laboratory. We thank Hui-Ting Chou and Zhiheng Yu at HHMI Janelia Research Campus, and Zheng Liu at NYU for assistance in microscope operation and data collection. For EM data processing, this work has utilized computing resources at the High-Performance Computing Facility at NYU Langone Medical Center. We thank Martin Ossowski and his HPC team as well as Joe Katsnelson for EM data processing support. The authors gratefully acknowledge funding support from the NIH National Institute of General Medical Sciences: R00GM112982 (G.B.), R01GM097312 (R.D.V.), Damon Runyon Cancer Research Foundation DFS-20-16 (G.B.), Howard Hughes Medical Institute (R.D.V.), and the UCSF Discovery Fellowship (S.N.). The study received financial support from the following: NIH National Institute of General Medical Sciences (R00GM112982), NIH National Institute of General Medical Sciences (R01GM097312), Damon Runyon Cancer Research Foundation DFS-20-16, Howard Hughes Medical Institute, Simons Foundation (349247), and NIH National Institute of General Medical Sciences (GM103310).

## Author contributions

GB, SN, and RDV conceived the research. GB, SN, NC, and NZ performed experiments and collected data. GB, SN, NC, and RDV analyzed data. GB, SN, and RDV wrote the manuscript. All authors edited the manuscript.

## Conflict of interest

The authors declare that they have no conflict of interest.

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
