## [Review Process File · The EMBO Journal]

Coupling of ATPase activity, microtubule binding and mechanics in the dynein motor domain

Stefan Niekamp, Nicolas Coudray, Nan Zhang, Ronald D. Vale & Gira Bhabha

Review timeline:

Submission date:	23rd Dec 2018
Editorial Decision:	15th Feb 2019
Revision received:	25th Mar 2019
Editorial Decision:	16th Apr 2019
Revision received:	24th Apr 2019
Accepted:	30th Apr 2019

Editor: Ieva Gailite

Transaction Report:

1st Editorial Decision

15th Feb 2019

Thank you for submitting your manuscript for consideration by The EMBO Journal. We have now received three referee reports on your manuscript, which are included below for your information.

As you will see from the reports, reviewers #1 and #3 express interest in the work and the proposed mechanism of stalk-mediated regulation of dynein activity, and all reviewers appreciate the quality of the data. However, reviewers #1 and #3 also raise a number of substantial concerns that would need to be addressed before they can support publication of the manuscript.

From my side, I judge the referee comments to be generally reasonable, therefore I would like to invite you to submit a revised manuscript addressing the concerns of reviewers #1 and #3. However, please note that it is The EMBO Journal policy to allow only a single major round of revision and that it is therefore important to resolve the main concerns at this stage.

We generally allow three months as standard revision time at the first instance. Please contact us in advance if you would need an additional extension. As a matter of policy, competing manuscripts published during this period will not negatively impact on our assessment of the conceptual advance presented by your study ("scooping" protection). However, please contact me as soon as possible upon publication of any related work in order to discuss how to proceed.

When preparing your letter of response to the referees' comments, please bear in mind that this will form part of the Review Process File, and will therefore be available online to the community. For more details on our Transparent Editorial Process, please visit our website:
http://emboj.embopress.org/about#Transparent_Process

REFeree REPORTS:

Referee #1:

In this study, the authors addressed the long-standing key question of how dynein, a huge microtubule-based molecular motor, carries out long-range two-way intramolecular communication to couple ATPase and microtubule-binding activities. They identified interesting mutations in dynein's stalk that connects the microtubule-binding domain and ATP-hydrolyzing AAA ring. The mutations trap dynein in nucleotide-independent weak binding to microtubules and microtubule-independent hyperactive ATPase state, indicating that the stalk-mediated two-way communication was disrupted in the mutants. By employing cryo-EM and single-particle analysis, the authors found that linker swing motions, which are believed to be the major contributor to dynein's force generation, would be inhibited in one of the mutants. They also found that nucleotide-dependent domain movements occur normally in one half of the ATP-hydrolyzing ring, but those in the other half of the ring are altered in the mutant. Based on these findings, the authors proposed a new working hypothesis of how the stalk apparatus controls structural changes in the ATP-hydrolyzing ring during dynein's mechanochemical cycle.

This is a well-written article containing very interesting results which merit publication in this journal. For the benefit of the readers, however, several points need clarifying and certain statements require further justification. These are given below.

Major concerns:

1. If feasible, the authors should show experimental evidence that the mutant-5 is indeed trapped in the ADP.Vi-bound state in the presence of ATP+Vi, which would be important information to justify the conclusion of this article. The mutant-5 is hyperactive and showed a "basal (no-MT)" ATPase rate that is as high as the maximal "MT-stimulated" ATPase rate in the wild-type enzyme. This raises concern that the structure of main ATPase site (at AAA1/AAA2) in the mutant could be different from that in the wild-type enzyme and that unlike the wild-type enzyme, the mutant could not be trapped in the ADP.Vi-bound state even in the presence of ATP+Vi. UV-vanadate photocleavage experiments and/or ATPase measurements of the mutant in the presence of ATP+Vi would provide such experimental evidence.

2. Please describe "diffusive-like" movements of mutant-2, mutant-5 and mutant-13 in more detail. Is this diffusion, stepping+diffusion, or biased diffusion? Quantitative information regarding the diffusive-like movements of the mutants along a MT track would be helpful for the readers to understand the characteristics of these interesting mutants.

3. The referee encourages the authors to reconsider the description of AMPPNP-bound state. In page 8 and in Supplementary Note 3, the authors stated "AMPPNP to mimic an ATP-bound state of the enzyme at AAA1 and AAA3". However, this expression would confuse the readers. As shown in the authors' previous study (Bhabha et al., 2014 Cell), all four nucleotide-binding sites (AAA1-AAA4) are occupied in the presence of AMPPNP. Furthermore, in general AMPPNP acts as an ATP analog, but for dynein, AMPPNP does not elicit ATP-induced key actions (detachment from MT/structural changes to the pre-force-generating state). Therefore, other expressions, e.g., "AMPPNP to mimic one of the post-force-generating states" would be better.

4. If possible, the referee would like the authors to discuss why the three mutants, mutant-2, mutant-5 and mutant-13, can drive directional MT sliding in the *in vitro* gliding assay because two critical actions of dynein's motility, that are, ATPase-induced affinity change for microtubules and ATPase-induced linker movements appear to be inhibited completely in the mutants.

Minor concerns and suggestions:

1. Page 4, line 3 from the top, "After ATP hydrolysis" should read "After ATP hydrolysis and product (phosphate) release". Previous biochemical studies suggest that dynein rebinds MTs not after ATP hydrolysis but after or during product releasing step.

2. In Fig 1B and in Supplementary Fig 4, figure legends and/or text in the figures appear to be incorrect. In Fig 1B, "mutant 15 was not expressed", but in Supplementary Fig 4A, the mutant-15 band is clearly visible; likewise, In the figure legend for Supplementary Fig 4A, "No dynein band is visible for mutant 15 and 16", but in Supplementary Fig 4A, no dynein band is visible for mutant 16 and 17.

3. Page 6, line 10 from the bottom, in Supplementary Fig 1L and in Supplementary Fig 3, it would be better to indicate the location and/or AA number of the stalk-buttruss interface because this is the key element for the conclusion of this study, and it would be useful for the readers to understand the structural features of the mutants described in this article.

4. Page 7, line 13 from the top, the main text describes "monomeric dynein", but in Supplementary Fig 6A, the cartoon would indicate that "dimeric dynein" was used in the MT-gliding assay.

5. Page 8, line 5 from the top, "for tubulin" should read "for tubulin dimer".

6. In Materials and Methods and in Supplementary Table 2, which describe microtubule affinity measurements, the two variables, k_{obs} and k_{basal} , should be defined. In addition, are B_M (maximum binding) values for the mutants OK? Supplementary Table 2 indicates that these values are around 0.1–0.3, suggesting that only a small fraction (10–30%) of the mutant molecules can bind MT even in the presence of infinite concentration of MT.

Referee #2:

In this study Niekamp et al. investigate the role of the dynein stalk region in linking ATPase activity and microtubule binding. Based upon extensive sequence analysis, they design a series of stalk mutants of the dynein motor domain and characterise their behavior in single molecule motility assays. They focus on three mutants that exhibit 'diffusive like' properties, in contrast to directional motility in the wildtype proteins. Further analysis of these mutants reveals that this diffusive behavior is nucleotide independent, and that they have lower affinity for microtubules than wildtype proteins. Furthermore, these mutants appear to uncouple ATPase activity from microtubule binding. To provide mechanistic understanding for these phenotypes, the authors employ single particle cryo-EM analysis to study one of these mutants (mutant 5) and conclude that this mutant affects nucleotide dependent changes in the AAA domain and linker - specifically in coupling changes in one half of the AAA ring with the other half.

Overall, the study appears robust and the conclusions drawn are appropriate. The discussion is quite speculative, but description of the results is appropriately restrained. However, I would question whether the extent of new insight from the single mutant studied in detail here really reaches the general significance for a wider readership requested by EMBO J.

Referee #3:

Niekamp et al. is an interdisciplinary study on cytoplasmic dynein, an ATP-driven motor protein. Dynein is a gigantic motor, consists of six AAA domains plus a long N-terminal tail with a helix-rich linker to AAA1, and a microtubule binding domain (MTBD). The AAA ring and MTBD are linked by a coiled-coil linker projecting from AAA5 and an additional coiled-coil called buttress protrudes from AAA4 and extends toward the linker. Therefore motions of individual parts at various nucleotide/force-generation conditions are key to understand the mechanism of dynein. A potential model of power stroke caused by conformational change of the AAA ring concomitant of ATPase and subsequent reorientation of the linker was proposed based on X-ray crystallography and cryo-EM structures from Burgess-, Vale-, Carter- and Kon-labs. The currently most potential model is that the AAA ring behaves separately between two half rings, one AAA2-AAA5 and the other AAA6 and AAA1 and that distance between them changes depending on the nucleotide status of AAA1 and AAA3. In addition, functional studies on dyneins genetically engineered at the coiled-coil stalk suggested conformational change of this region evokes coupling between ATPase and microtubule binding. However there has been no study to combine genetic engineering of the stalk and high resolution structural work to elucidate the mechanism of this coupling at atomic level. To address this question, the author engineered 18 mutants, targeting the stalk. They especially focused on one mutant (mutant 5), which interestingly shows diffusive motion on the microtubule (even without ATP) and higher basal ATPase, but complete lack of its activation by microtubules. They reconstructed 3D structure of this mutant dynein by cryo-EM in the presence of AMPPNP (mimicking the ATP state) and in the presence of ADP.Vi (mimicking ADP.Pi). With help of improved spatial resolution, they demonstrated precisely that the conformational change from AMPPNP to ADP.Vi within the AAA ring occurs between a half ring from the small subunit of AAA2 to the large subunit of AAA5 and the other half of the ring (the small subunit of AAA5 to the

large subunit of AAA2) during transition from AMPPNP to ADP.Vi. This motion is significantly suppressed by mutation 5, which indicates that conformational change in AAA1 induced by ATP hydrolysis is propagated to the other half of the ring through the stalk (and probably the buttress), suggesting mutual influence between the stalk/MTBD and the two half AAA rings during the ATPase cycle. They attempted to extend their discussion on general mechanism how ATPase and MT binding are coupled.

The experimental facts they found are sound and fascinating. Both function and structure of this mutant provides insights into the dynein mechanism and will evoke discussion in this field. Their structure at high resolution clarified which parts of dynein are essential to move between nucleotide conditions. This reviewer, however, has a few concerns in their interpretation and hypotheses. Hopefully the authors can address the following points and revise the manuscript to a publishable form in EMBO Journal.

Major concerns:

1. Missing structure: The authors reconstructed a 9.2Å resolution apo structure of mutant 5, according to Methods (p.20), but did not show it in the manuscript. The apo structure will help us to understand the ATPase cycle of this mutant to draw a diagram like Fig.7 of Bhabha et al. 2014 and helps discussion below. This reviewer strongly encourages the authors to present apo structure.

2. For this reviewer, their discussion how mutant 5 has hyperactive basal ATPase is not clear. According to the discussion of this group in the past (Fig. 7 of Bhabha et al. 2014), AAA1 and AAA2 must be close to each other at ADP.Pi state to facilitate ATPase. However mutant 5 seems to have rather an open ring conformation - the distance between the AAA2L insert loop and AAA1L in mutant 5 with ADP.Vi is shorter than that of AMPPNP, but longer than WT with ADP.Vi (by the way, graphic presentation in Fig.4B and Sup Fig.7FG could be improved to help us understand relative positioning of AAA1 and AAA2 in the context of ATP hydrolysis, maybe by showing AAA1L, AAA1S, AAA2L and possible ATP location at the same time and in stereo or movie), suggesting transition from ADP.Pi to ADP (Pi release, which is the rate-determining step) of mutant 5 is slower than WT. How can we explain faster ATPase cycle in mutant 5?

3. The authors emphasized that the linker takes straight form, instead of a bent form, with ADP.Vi. It elegantly explains lack of power stroke, including diffusive motion and missing ATPase activation by microtubule binding. However, they further speculate that this abnormal linker conformation influences ATPase (and in WT at ADP.Pi steric clash of the linker is essential for ATPase cycle). Do they claim the bent linker plays a regulatory role to slow down Pi release in WT (and mutant 5 excludes this suppression)? They could at least model the AAA-ring of mutant 5 and WT at ADP.Vi with the linker at straight position to prove it causes clash in WT, but not in mutant 5.

4. The authors started the study with 18 mutants (all targeting the stalk), but pursued three mutants with diffusive motion for biochemical characterization and finally only mutant 5 for further structural analysis. How do they assess other mutants in the context of their hypothesis? They found fewer CC1 mutations affect functions than CC2 mutation. Otherwise there are not many statements for other mutants. Since the position of mutation in mutant 5 is not the stalk/buttress joint, we cannot highlight mutant 5 as the proof of importance of buttress/stalk interaction. How about other mutants? Are there any (except three diffusive ones) mutants which show hyper basal ATPase? Or do only diffusive dyneins show hyperactive ATPase? Ideally discussion how mutation at the particular locus influences (or does not influence) ATPase and power stroke in each mutant will make more sense to present 18 mutants.

Minor points:

5. They claim 7.6-7.7Å resolution from single particle analysis. While this should be enough resolution to identify secondary structure, Figs. 3A, 4AC, Supplementary Fig. 7DE do not seem so high resolution (Supplementary Fig. 7C seems reasonable), although current resolution is enough to discuss subdomain motion. Is there significant bias of spatial resolution within the map?

6. The way to mention mutants in the text can be improved. For example, they do not mention the numbering of mutants explicitly in p.6 and therefore it is hard to guess which mutant they have in mind as mutation at the interface of stalk/buttress.

Response to Referees

Referee comments are in black, and our responses are in blue. All changes in the manuscript are in red.

Referee #1:

In this study, the authors addressed the long-standing key question of how dynein, a huge microtubule-based molecular motor, carries out long-range two-way intramolecular communication to couple ATPase and microtubule-binding activities. They identified interesting mutations in dynein's stalk that connects the microtubule-binding domain and ATP-hydrolyzing AAA ring. The mutations trap dynein in nucleotide-independent weak binding to microtubules and microtubule-independent hyperactive ATPase state, indicating that the stalk-mediated two-way communication was disrupted in the mutants. By employing cryo-EM and single-particle analysis, the authors found that linker swing motions, which are believed to be the major contributor to dynein's force generation, would be inhibited in one of the mutants. They also found that nucleotide-dependent domain movements occur normally in one half of the ATP-hydrolyzing ring, but those in the other half of the ring are altered in the mutant. Based on these findings, the authors proposed a new working hypothesis of how the stalk apparatus controls structural changes in the ATP-hydrolyzing ring during dynein's mechanochemical cycle.

This is a well-written article containing very interesting results which merit publication in this journal. For the benefit of the readers, however, several points need clarifying and certain statements require further justification. These are given below.

Major concerns:

1. If feasible, the authors should show experimental evidence that the mutant-5 is indeed trapped in the ADP.Vi-bound state in the presence of ATP+Vi, which would be important information to justify the conclusion of this article. The mutant-5 is hyperactive and showed a "basal (no-MT)" ATPase rate that is as high as the maximal "MT-stimulated" ATPase rate in the wild-type enzyme. This raises concern that the structure of main ATPase site (at AAA1/AAA2) in the mutant could be different from that in the wild-type enzyme and that unlike the wild-type enzyme, the mutant could not be trapped in the ADP.Vi-bound state even in the presence of ATP+Vi. UV-vanadate photo-cleavage experiments and/or ATPase measurements of the mutant in the presence of ATP+Vi would provide such experimental evidence.

We agree that information about the presence or absence of ATP-Vi in the AAA1 nucleotide binding pocket is important. Therefore we performed an ATPase assay in the presence of ATP-Vi for wild-type and mutant 5. We found that for both, wild-type and

mutant 5 the ATPase activity is inhibited in the presence of vanadate, showing that these proteins both respond similarly to vanadate inhibition. In addition, we conducted the UV-vanadate photo-cleavage experiment as suggested for mutant 5 and saw two bands at ~270 kDa and ~90 kDa. Taken together, these experiments suggested by the reviewer indicate that vanadate is present in the AAA1 nucleotide binding pocket. For both experiments we used the same protein preparation that was used to determine the structure of mutant 5 in the presence of ATP-Vi. These results support that mutant 5 is trapped in the ATP-Vi state under the conditions used for structure determination. The results are presented in Figure EV5F-H (former Supplementary Fig. 9) and described on page 11.

2. Please describe "diffusive-like" movements of mutant-2, mutant-5 and mutant-13 in more detail. Is this diffusion, stepping+diffusion, or biased diffusion? Quantitative information regarding the diffusive-like movements of the mutants along a MT track would be helpful for the readers to understand the characteristics of these interesting mutants.

This is an interesting question and we addressed it by measuring the displacement distance and directionality per one second interval of mutant 5 and wild-type dynein. We tracked single molecules of wild-type dynein and mutant 5 along microtubules in the presence of 1 mM ATP. We then straightened these tracks along the main axis of motion along the microtubule using the 'localization microscopy' plug-in from μ Manager 2.0 (Edelstein *et al*, 2010). Afterwards, the displacements of wild-type and mutant 5 were binned into 1 sec intervals and the polarity of microtubules was determined by analyzing the directionality of human homodimeric kinesin-1 (K490) (Tomishige *et al*, 2006), which moves processively towards the plus end of microtubules. The histogram of the displacements for mutant 5 (Appendix Fig. S5) reveals a uniform Gaussian distribution centered close to zero with an average displacement of -3.3 nm. This analysis supported the notion that the back-and-forth motion of mutant 5 reflects random thermal-driven motion along the microtubule. The data is presented in Appendix Figure S5 and we added a few sentences in the manuscript on page 7.

3. The referee encourages the authors to reconsider the description of AMPPNP-bound state. In page 8 and in Supplementary Note 3, the authors stated "AMPPNP to mimic an ATP-bound state of the enzyme at AAA1 and AAA3". However, this expression would confuse the readers. As shown in the authors' previous study (Bhabha *et al.*, 2014 *Cell*), all four nucleotide-binding sites (AAA1-AAA4) are occupied in the presence of AMPPNP. Furthermore, in general AMPPNP acts as an ATP analog, but for dynein, AMPPNP does not elicit ATP-induced key actions (detachment from MT/structural changes to the pre-force-generating state). Therefore, other expressions, e.g., "AMPPNP to mimic one of the post-force-generating states" would be better.

That is a great point. We changed it accordingly on page 9 and in Appendix Note S3.

4. If possible, the referee would like the authors to discuss why the three mutants, mutant-2, mutant-5 and mutant-13, can drive directional MT sliding in the in vitro gliding assay because two critical actions of dynein's motility, that are, ATPase-induced affinity change for microtubules and ATPase-induced linker movements appear to be inhibited completely in the mutants.

This is indeed an interesting point, and not unique to these mutants. For example, human cytoplasmic dynein (dynein 1) purified from rat brains also has been shown to be diffusive in single-molecule assays (McKenney *et al*, 2014), while showing directional motility in gliding assays. While our mutants can translocate microtubules in a directional manner, the gliding velocity is reduced by about 10 fold compared to wild-type dynein (Appendix Fig. S6). The difference between the single-molecule and multiple motor gliding assay might reflect mechanical coupling or other types of cooperativity. However, pinpointing the mechanism is beyond the scope of this work and is not central to the main points of this study. Nevertheless, we have included a few sentences on page 8 to highlight this observation.

Minor concerns and suggestions:

1. Page 4, line 3 from the top, "After ATP hydrolysis" should read "After ATP hydrolysis and product (phosphate) release". Previous biochemical studies suggest that dynein rebinds MTs not after ATP hydrolysis but after or during product releasing step.

We thank the reviewer for pointing this out, and have changed this phrase accordingly in the revised manuscript on page 4.

2. In Fig 1B and in Supplementary Fig 4, figure legends and/or text in the figures appear to be incorrect. In Fig 1B, "mutant 15 was not expressed", but in Supplementary Fig 4A, the mutant-15 band is clearly visible; likewise, In the figure legend for Supplementary Fig 4A, "No dynein band is visible for mutant 15 and 16", but in Supplementary Fig 4A, no dynein band is visible for mutant 16 and 17.

We thank the reviewer for catching this mistake. We accidentally skipped over construct #10 in the gel in Figure EV2A (former Supplementary Figure 4) and listed 19 instead of 18 mutants. This has been corrected, and it is now consistent with Figure 1B.

3. Page 6, line 10 from the bottom, in Supplementary Fig 1L and in Supplementary Fig 3, it would be better to indicate the location and/or AA number of the stalk-buttruss interface because this is the key element for the conclusion of this study, and it would be useful for the readers to understand the structural features of the mutants described in this article.

This is a very good suggestion. We have added the AA number for the stalk / buttress interface in the revised version of Figure EV1A (former Supplementary Fig. 1). We now refer to this panel in the manuscript on page 6 as well. However, we did not add it in Appendix Figure S2 (former Supplementary Fig. 3) because the goal of this figure is to highlight the position of insertion or deletion of each mutant, and we felt that adding the stalk / buttress interface position in addition might be distracting.

4. Page 7, line 13 from the top, the main text describes "monomeric dynein", but in Supplementary Fig 6A, the cartoon would indicate that "dimeric dynein" was used in the MT-gliding assay.

We apologize for the confusion. The cartoon is correct as this assay uses dimeric dynein. We have changed the text in the manuscript accordingly. Moreover, we have also added this information to the Materials and Methods section.

5. Page 8, line 5 from the top, "for tubulin" should read "for tubulin dimer".

We have changed the wording accordingly (see page 9).

6. In Materials and Methods and in Supplementary Table 2, which describe microtubule affinity measurements, the two variables, k_{obs} and k_{basal} , should be defined. In addition, are B_m (maximum binding) values for the mutants OK? Supplementary Table 2 indicates that these values are around 0.1~0.3, suggesting that only a small fraction (10~30%) of the mutant molecules can bind MT even in the presence of infinite concentration of MT.

We agree that k_{obs} and k_{basal} should be defined, and we now define these values in the legend of Appendix Table S2 and in Materials and Methods on page 20. In the weak binding states of the motor, it is not technically feasible to obtain saturating concentrations of microtubules. Hence we agree that B_m values do not make sense. Consistent with similar experiments previously published in the literature (Imamura *et al*, 2007; Kon *et al*, 2009), we have now indicated these values as "not measurable" (n/m) for the weak binding states.

Referee #2:

In this study Niekamp et al. investigate the role of the dynein stalk region in linking ATPase activity and microtubule binding. Based upon extensive sequence analysis, they design a series of stalk mutants of the dynein motor domain and characterise their behavior in single molecule motility assays. They focus on three mutants that exhibit 'diffusive like' properties, in contrast to directional motility in the wildtype proteins. Further analysis of these mutants reveals that this diffusive behavior is nucleotide independent, and that they have lower affinity for microtubules than wildtype proteins. Furthermore, these mutants appear to uncouple ATPase activity from microtubule binding. To provide mechanistic understanding for these phenotypes, the authors employ single particle cryo-EM analysis to study one of these mutants (mutant 5) and conclude that this mutant affects nucleotide dependent changes in the AAA domain and linker - specifically in coupling changes in one half of the AAA ring with the other half.

Overall, the study appears robust and the conclusions drawn are appropriate. The discussion is quite speculative, but description of the results is appropriately restrained. However, I would question whether the extent of new insight from the single mutant studied in detail here really reaches the general significance for a wider readership requested by EMBO J.

This reviewer did not request specific revisions, therefore no changes have been made in response to the review.

Referee #3:

Niekamp et al. is an interdisciplinary study on cytoplasmic dynein, an ATP-driven motor protein. Dynein is a gigantic motor, consists of six AAA domains plus a long N-terminal tail with a helix-rich linker to AAA1, and a microtubule binding domain (MTBD). The AAA ring and MTBD are linked by a coiled-coil linker projecting from AAA5 and an additional coiled-coil called buttress protrudes from AAA4 and extends toward the linker. Therefore motions of individual parts at various nucleotide/force-generation conditions are key to understand the mechanism of dynein. A potential model of power stroke caused by conformational change of the AAA ring concomitant of ATPase and subsequent reorientation of the linker was proposed based on X-ray crystallography and cryo-EM structures from Burgess-, Vale-, Carter- and Kon-labs. The currently most potential model is that the AAA ring behaves separately between two half rings, one AAA2-AAA5 and the other AAA6 and AAA1 and that distance between them changes depending on the nucleotide status of AAA1 and AAA3. In addition, functional studies on dyneins genetically engineered at the coiled-coil stalk suggested conformational change of this region evokes coupling between ATPase and microtubule binding. However there has been no study to combine genetic engineering of the stalk and high resolution structural work to elucidate the mechanism of this coupling at atomic level.

To address this question, the author engineered 18 mutants, targeting the stalk. They especially focused on one mutant (mutant 5), which interestingly shows diffusive motion on the microtubule (even without ATP) and higher basal ATPase, but complete lack of its activation by microtubules. They reconstructed 3D structure of this mutant dynein by cryo-EM in the presence of AMPPNP (mimicking the ATP state) and in the presence of ADP.Vi (mimicking ADP.Pi). With help of improved spatial resolution, they demonstrated precisely that the conformational change from AMPPNP to ADP.Vi within the AAA ring occurs between a half ring from the small subunit of AAA2 to the large subunit of AAA5 and the other half of the ring (the small subunit of AAA5 to the large subunit of AAA2) during transition from AMPPNP to ADP.Vi. This motion is significantly suppressed by mutation 5, which indicates that conformational change in AAA1 induced by ATP hydrolysis is propagated to the other half of the ring through the stalk (and probably the buttress), suggesting mutual influence between the stalk/MTBD and the two half AAA rings during the ATPase cycle. They attempted to extend their discussion on general mechanism how ATPase and MT binding are coupled.

The experimental facts they found are sound and fascinating. Both function and structure of this mutant provides insights into the dynein mechanism and will evoke discussion in this field. Their structure at high resolution clarified which parts of dynein are essential to move between nucleotide conditions. This reviewer, however, has a few concerns in their interpretation and hypotheses. Hopefully the authors can address the following points and revise the manuscript to a publishable form in EMBO Journal.

Major concerns:

1. Missing structure: The authors reconstructed a 9.2Å resolution apo structure of mutant 5, according to Methods (p.20), but did not show it in the manuscript. The apo structure will help

us to understand the ATPase cycle of this mutant to draw a diagram like Fig.7 of Bhabha et al. 2014 and helps discussion below. This reviewer strongly encourages the authors to present apo structure.

This was an unfortunate typo and we apologize! It should have read “The model for the 9.2 Å map was constructed as for mutant 5 in AMPPNP-state, using rigid body docking of domains from PDB 4W8F, but was not further refined in PHENIX due to its lower resolution”. If we had the structure for the apo-state, we would certainly include it. We agree with the reviewer that it would be interesting to have the full cycle of the mutant. However, we feel that the apo structure is not required to validate the primary conclusions we made in this study.

2. For this reviewer, their discussion how mutant 5 has hyperactive basal ATPase is not clear. According to the discussion of this group in the past (Fig. 7 of Bhabha et al. 2014), AAA1 and AAA2 must be close to each other at ADP.Pi state to facilitate ATPase. However mutant 5 seems to have rather an open ring conformation - the distance between the AAA2L insert loop and AAA1L in mutant 5 with ADP.Vi is shorter than that of AMPPNP, but longer than WT with ADP.Vi (by the way, graphic presentation in Fig.4B and Sup Fig.7FG could be improved to help us understand relative positioning of AAA1 and AAA2 in the context of ATP hydrolysis, maybe by showing AAA1L, AAA1S, AAA2L and possible ATP location at the same time and in stereo or movie), suggesting transition from ADP.Pi to ADP (Pi release, which is the rate-determining step) of mutant 5 is slower than WT. How can we explain faster ATPase cycle in mutant 5?

We thank the reviewer for the comment and suggestions. In the revised manuscript, we compare the position of AAA1L, AAA1s, and AAA2L between wild-type ADP-vi and wild-type AMPPNP, between mutant 5 ADP-vi and mutant 5 AMPPNP, and between wild-type ADP-vi and mutant 5 ADP-vi (Fig. 4B). We also modeled a stick representation of ADP-vi based on the wild-type ADP-vi structure to all figures to orient the reader. Moreover, we now show the angle between AAA1L and AAA2L for different structures (Fig. EV5C, D) and the approximate distance between Walker-A motif and the Arginine finger (Fig. EV5E) for all four structures (wild-type AMPPNP, wild-type ADP-vi, mutant 5 AMPPNP, and mutant 5 ADP-vi), based on backbone placement. We have added a more detailed description on page 10 in the manuscript: “Based on fitting AAAs and AAAL domains into our density as described above, our data show that the gap between AAA1L and AAA2L for mutant 5 closes when transitioning from the AMPPNP to the ADP-vi state (Fig. 4B, Movie EV13), as observed for wild-type dynein. The AAA2L domain of mutant 5 undergoes a rotation between the AMPPNP and ADP-vi state of 21° which is similar to the AAA2L domain rotation of wild-type dynein with 20° (Fig. EV5C, D). Moreover, the distance between the Arginine finger and Walker-A of mutant 5 and wild-type change from ~22 Å and ~20 Å in the AMPPNP state, respectively to ~17 Å and ~14 Å in the ADP-vi state, respectively (Fig. EV5E), highlighting that the gap between AAA1L and AAA2L for mutant 5 and wild-type dynein indeed close in a similar manner.” Thus, mutant 5 can likely hydrolyze ATP with a

similar efficiency as wild-type dynein but without the regulation of microtubules, which slows the ATP hydrolysis of wild-type dynein. We also discuss this observation on page 14: “Specifically, the disruption of the stalk-buttruss interface allows the buttruss and AAA5s/AAA6/AAA1L to undergo open-closed transitions accompanying ATP binding, hydrolysis and product release, without any regulation by microtubules through the stalk-buttruss apparatus.”

3. The authors emphasized that the linker takes straight form, instead of a bent form, with ADP.Vi. It elegantly explains lack of power stroke, including diffusive motion and missing ATPase activation by microtubule binding. However, they further speculate that this abnormal linker conformation influences ATPase (and in WT at ADP.Pi steric clash of the linker is essential for ATPase cycle). Do they claim the bent linker plays a regulatory role to slow down Pi release in WT (and mutant 5 excludes this suppression)? They could at least model the AAA-ring of mutant 5 and WT at ADP.Vi with the linker at straight position to prove it causes clash in WT, but not in mutant 5.

We apologize for the lack of clarity in this explanation. We do not claim that the bent linker plays a regulatory role to slow down Pi release in wild-type. We speculate that one explanation for why the linker does not adopt a bent conformation in mutant 5 is that AAA5L does not undergo the “normal” conformational change as it does in the wild-type enzyme. This is highlighted in the “gap” observed between AAA5L and AAA5s. In the wild-type enzyme, the AAA5L conformational change in the presence of ATP-Vi would lead to a steric clash if the linker did not bend. In the mutant, this conformational change is not observed for AAA5L, therefore there is no steric clash with the linker, and therefore the linker bending may not be initiated. We have reworked this explanation in the discussion on page 15. And as suggested by the reviewer, Figure EV5K has been added to highlight the steric clash between AAA5L and the linker in the wild-type protein, but not in mutant 5.

4. The authors started the study with 18 mutants (all targeting the stalk), but pursued three mutants with diffusive motion for biochemical characterization and finally only mutant 5 for further structural analysis. How do they assess other mutants in the context of their hypothesis? They found fewer CC1 mutations affect functions than CC2 mutation. Otherwise there are not many statements for other mutants. Since the position of mutation in mutant 5 is not the stalk/buttruss joint, we cannot highlight mutant 5 as the proof of importance of buttruss/stalk interaction. How about other mutants? Are there any (except three diffusive ones) mutants which show hyper basal ATPase? Or do only diffusive dyneins show hyperactive ATPase? Ideally discussion how mutation at the particular locus influences (or does not influence) ATPase and power stroke in each mutant will make more sense to present 18 mutants.

Since the position of mutation in mutant 5 is not the stalk/buttress joint, we cannot highlight mutant 5 as the proof of importance of buttress/stalk interaction. How about other mutants?

We thank the reviewer for suggesting that we highlight mutants that may speak to the importance of the stalk-buttress interaction. Most of the mutations that resulted in a dead (mutants 8, 9, 10, 17, and 18) or unstable (mutants 15 and 16) motor are clustered in the proximal region of the stalk, close to the AAA ring (Fig. EV1L), and are in regions that are important for the stalk/buttress interaction (Fig. EV1A, L, Appendix Fig. S2). Of these, mutants 15 and 16 could not be expressed recombinantly, suggesting that these affect overall stability of the protein. Mutants 8, 9, 10, 17 and 18 could be expressed and purified, but showed no movement in single-molecule motility assays (Fig. 1B, Fig. EV2C), suggesting that mutations in these regions severely compromise dynein motility. However, some 'wild-type' like mutants (mutants 1, 3, 11, and 12) also have insertions or deletions in the vicinity of the stalk / buttress interface but do not seem to impact motility. These mutations are all in CC1 indicating that CC2 might be more sensitive to insertions or deletions in the stalk / buttress interface. One potentially explanation for this observation is that the insertions or deletions in CC2 disturb the previously described kinking of CC2 upon ATP hydrolysis (Schmidt *et al*, 2015) and are therefore more deleterious. While the mechanistic basis of this observation would require extensive structural characterization of each mutant, and is beyond the scope of this study, these results do support that the stalk-buttress interaction is important, and we have now included this on page 6 / 7 of the main text and in Appendix Note S2.

How do they assess other mutants in the context of their hypothesis? Are there any (except three diffusive ones) mutants which show hyper basal ATPase? Or do only diffusive dyneins show hyperactive ATPase?

We would like to thank the reviewer for encouraging us to revisit data for all our mutants, and as much as possible, we have included data and interpretation for these in the revised manuscript.

We observed one mutant (mutant 14) that has a high basal ATPase activity which is also independent of microtubules concentration, but does not show diffusive movement. This mutant contains two distinct populations of molecules: the majority (96%) transiently binds to and releases from microtubules and the minor population (4%) appears to move in a similar fashion as the wild-type motor (Fig. EV3A, B). Because the percentage of moving motors is small, to rule out contamination of mutant 14 with wild-type enzyme, we repeated preps of mutant 14 using size exclusion columns that had never been used to purify wild-type dynein and the result was reproducible. In the absence of ATP, we observe transient binding events, which suggest that mutant 14 on average has weak affinity for microtubules in the apo state, opposite to the wild-type protein. A mechanistic understanding of this mutant will surely be interesting but is beyond the scope of this work as it would likely require high resolution structural

information in multiple states, and the ability to capture snapshots of both populations in multiple states. Initially, we only analyzed moving motors for mutant 14, and hence listed it as “wild-type like”. In the revised version of the manuscript, we have described the single-molecule and ATPase assays in more depth and thoroughly. Interestingly, the site of mutation for mutant 14 overlaps with the mutation site of mutant 18, yet mutant 18 does not show any motility. These mutants will be worth considering for in-depth future studies. For mutant 14 we also performed single-molecule assays with and without ATP and measured the microtubule affinity by a cosedimentation assay. The data of these experiments is presented in an additional Supplementary Figure (Fig. EV3). We have added all this information about mutant 14 on page 6-9. Moreover, we modified Figure 1 and Figure EV1 and EV2 with the new classification of mutant 14 as ‘transient binding’.

Mutants 1, 3, 4, 6, 7, 11, and 12 show very similar motility to wild type, suggesting that the regions of these mutations are more tolerant to changes in length and sequence. We have now noted this in the results section on page 6 and 7.

Further characterization of some of these mutants by us or others will be interesting but is beyond the scope of this study.

Minor points:

5. They claim 7.6-7.7A resolution from single particle analysis. While this should be enough resolution to identify secondary structure, Figs. 3A, 4AC, Supplementary Fig. 7DE do not seem so high resolution (Supplementary Fig. 7C seems reasonable), although current resolution is enough to discuss subdomain motion. Is there significant bias of spatial resolution within the map?

An additional Supplementary Figure (Appendix Fig. S7) was added to show the local resolution of the two maps. While the overall resolution of both maps is quite similar (~7.6-7.7A), we indeed notice heterogeneities that are highlighted by the heatmaps. More specifically, the range of resolution is larger for class 2, with the best regions reaching a resolution of 7A, at which the secondary structure is clearly resolved (AAA4, AAA5 and the linker), while a large part of AAA1 and AAA2 are at lower resolution, where secondary structure is not decipherable. The resolution within class 1 is more homogeneous, but still displays slightly better resolution in similar locations. The resolutions we report are the average resolution after refinement in CryoSparc v2.5.0. This has been made clear in the revised version of the Materials and Methods section of the manuscript.

6. The way to mention mutants in the text can be improved. For example, they do not mention the numbering of mutants explicitly in p.6 and therefore it is hard to guess which mutant they have in mind as mutation at the interface of stalk/buttness.

We agree. In the revised manuscript, we have specified mutant numbers from the beginning, on page 6, and throughout the manuscript. We have also made this more clear in the Appendix Note S2 where we discuss phenotypes of various mutants in more detail.

References

- Edelstein A, Amodaj N, Hoover K, Vale R & Stuurman N (2010) Computer control of microscopes using μ Manager. *Curr. Protoc. Mol. Biol.* **Chapter 14:** Unit14.20
- Imamura K, Kon T, Ohkura R & Sutoh K (2007) The coordination of cyclic microtubule association/dissociation and tail swing of cytoplasmic dynein. *Proc. Natl. Acad. Sci. U. S. A.* **104:** 16134–16139
- Kon T, Imamura K, Roberts AJ, Ohkura R, Knight PJ, Gibbons IR, Burgess SA & Sutoh K (2009) Helix sliding in the stalk coiled coil of dynein couples ATPase and microtubule binding. *Nat. Struct. Mol. Biol.* **16:** 325–333
- McKenney RJ, Huynh W, Tanenbaum ME, Bhabha G & Vale RD (2014) Activation of cytoplasmic dynein motility by dynactin-cargo adapter complexes. *Science* **345:** 337–341
- Schmidt H, Zalyte R, Urnavicius L & Carter AP (2015) Structure of human cytoplasmic dynein-2 primed for its power stroke. *Nature* **518:** 435–438
- Tomishige M, Stuurman N & Vale RD (2006) Single-molecule observations of neck linker conformational changes in the kinesin motor protein. *Nat. Struct. Mol. Biol.* **13:** 887–894

2nd Editorial Decision

16th Apr 2019

Thank you for submitting a revised version of your manuscript. It has now been seen by two of the original referees, who find that their main concerns have been addressed and are now in favour of publication of the manuscript. There remain only a few editorial issues that have to be dealt with before I can extend formal acceptance of the manuscript:

REFeree REPORTS:

Referee #1:

In this revised manuscript, the authors have performed additional experiments and have addressed most of the concerns raised by the reviewer. Now this manuscript would be acceptable for publication.

Referee #3:

The authors addressed all the points this reviewer raised. This reviewer is convinced with the revised manuscript and thinks that they also adequately dealt with the points from the other reviewers. This reviewer believes, although they mainly focus on one mutant, the revised version describes biophysical and biochemical characteristics of other mutants as well, which is systematic and informative. Taken together I recommend publication of this manuscript in the EMBO Journal.

2nd Revision - authors' response

24th Apr 2019

The authors performed all requested editorial changes.

Corresponding Author Name: Gira Bhabha

Manuscript Number: EMBOJ-2018-101414